# Performance Enhancement of Internal Combustion Engines through Vibration Control: State of the Art and Challenges

**Hojat Mahdisoozani [1], Mehrdad Mohsenizadeh [2], Mehdi Bahiraei [3], Alibakhsh Kasaeian [4], Armin Daneshvar [5], Marjan Goodarzi [6] and Mohammad Reza Safaei [7,8,*]**

[1] Department of Mechanical Engineering, Iran University of Science and Technology, Tehran 16844, Iran; hojat_mahdisoozani@mecheng.iust.ac.ir

[2] Mechanical Engineering Department, Lamar University, Beaumont, Texas 77706, USA; mmohsenizade@lamar.edu

[3] Department of Mechanical Engineering, Kermanshah University of Technology, Kermanshah 6715685420, Iran; m.bahiraei@kut.ac.ir

[4] Faculty of New Sciences and Technologies, University of Tehran, Tehran 1417466191, Iran; akasa@ut.ac.ir

[5] Mechanical Engineering Department, Lamar University, Beaumont, Texas 77706, USA; armin.daneshvar@gmail.com

[6] Department of Mechanical Engineering, Lamar University, Beaumont, Texas 77705, USA; mgoodarzi@lamar.edu

[7] Division of Computational Physics, Institute for Computational Science, Ton Duc Thang University, Ho Chi Minh City 756604, Vietnam

[8] Faculty of Electrical and Electronics Engineering, Ton Duc Thang University, Ho Chi Minh City 756604, Vietnam

**\*** Correspondence: cfd_safaei@tdtu.edu.vn; Tel.: +1-502-657-9981

**Abstract:** Internal combustion engines (ICEs) are the primary source of power generation in today's driving vehicles. They convert the chemical energy of the fuel into the mechanical energy which is used to drive the vehicle. In this process of energy conversion, several parameters cause the engine to vibrate, which significantly deteriorate the efficiency and service life of the engine. The present study aims to gather all the recent works conducted to reduce and isolate engine vibration, before transmitting to other vehicle parts such as drive shafts and chassis. For this purpose, a background history of the ICEs, as well as the parameters associated with their vibration, will be introduced. The body of the paper is divided into three main parts: First, a brief summary of the vibration theory in fault detection of ICEs is provided. Then, vibration reduction using various mechanisms and engine modifications is reviewed. Next, the effect of using different biofuels and fuel additives, such as alcohols and hydrogen, is discussed. Finally, the paper ends with a conclusion, summarizing the most recent methods and approaches that studied the vibration and noise in the ICEs.

**Keywords:** internal combustion engine; vibration; noise; NVH; biodiesel; direct injection; emission; hydrogen; short-time Fourier transform; kurtosis; Morlet wavelet

## 1. Introduction

An internal combustion engine (ICE) is powered by conversion of released energy from an air/fuel mixture, generated in fixed cylinders of the ICE, to the mechanical energy of rotating crankshafts. Due to parameters, such as unbalanced reciprocating and rotating parts, cyclic variations in gas pressure generated by the combustion process, misfire, and inertia forces of the

reciprocating parts, significant levels of vibration are induced in ICEs. The vibration signals are categorized as torsional, longitudinal, and mixed vibrations. Torsional vibrations are mainly caused by the exertion of cyclic combustion forces within the cylinder as well as the inertial forces of rotating parts, such as the crankshaft, camshaft, and connecting rod. The primary source of longitudinal vibrations is the unbalanced forces acting on reciprocating and rotating components of the engine which propagate in three orthogonal directions [1]. The interactions of both longitudinal and torsional vibrations are called mixed vibrations, which is the case in all ICEs.

The internal combustion vehicles (ICVs) suffer from low transmission efficiency, lubrication faults, wear, fatigue, scuffing faults, and poor drivability, caused by the engine vibrations. The inertial forces of rotating components, as well as harmonic combustion forces in the cylinders, eccentric rotation of journal bearing support and flywheel, and transient contact dynamics of the cam/follower, are other contributions to the vibration problem in ICVs. Noises generated by these vibrations are transmitted through the engine mounts and chassis to backrests, which can endanger passengers' life and safety of the vehicle. The level of safety and vibration isolation depends on the amplitude, wave shape, and duration of exposure to these noises [2]. Vibration noises are products of the motion generated at the engine that originate from the fuel energy. Hence, a ratio of the fuel energy that can be used for producing useful work is lost due to the vibrations.

In this review paper, different types of vibration in ICEs and their fundamental sources are discussed. Furthermore, it is structured to provide scientific and experimentally based methods through which the vibration can be attenuated efficiently. These vibration reduction methods are categorized into two types: (i) applying engine modifications by changing the dimensions and geometry of the engine parts and designing novel mechanisms to isolate the engine vibrations; and (ii) adding of additives such as biofuels, hydrogen gas, metallic nanoparticles, and alcohols to the conventional diesel fuel.

## 2. Measurement of Torsional Vibrations in ICEs

Early fault detection and diagnosis for diesel engines is essential to ensure reliable operation during their service life. The reciprocating motion of ICEs always induces some degree of torsional vibration during their operation. The engine piston experiences different strikes with the rotation of the crankshaft. Any increase in the rotation speed of the crankshaft leads to a pressure rise in the cylinder as the piston reaches the top dead center (TDC) on the compression stroke. The pressure is further increased at the ignition and combustion stages, immediately after TDC. This pressure is released as soon as gases are expanded when the piston approaches bottom dead center (BDC). The pressure is transferred to the piston through the connecting rod, generating tangential forces that are useful work. While the useful work increases, the rotation speed of the crankshaft augments during the combustion stroke. That is while, the compression stroke, in turn, reduces engine's angular velocity. These fluctuations in the crankshaft speed make the detection of the compression stroke of the firing cylinder very difficult [3].

For successful fault detection, it is vital to model the dynamics of the engine components, especially the torsional vibration of the crankshaft. Lagrange rule is a reliable method for such modeling [3]:

$$\frac{d}{dt}\left(\frac{\partial T}{\partial \dot{\theta}_i}\right) - \frac{\partial T}{\partial \theta_i} + \frac{\partial U}{\partial \theta_i} + \frac{\partial D}{\partial \theta_i} = Q_i, \ i = 0, 1, \dots, n, \tag{1}$$

where $T$ is the kinetic energy and $U$ is the potential energy of the crankshaft, defined based on degrees of freedom (df) of the diesel engine. Also, $D$ shows the dissipative energy which depends on the inertia, torsional stiffness, damping, and number of mass, as well as angular displacement of each mass.

In real-world applications, engines are subjected to numerous external excitations during their operation. An external excitation torque of a single cylinder includes both the combustion gas torque $(M_p)$ and reciprocating inertia torque $(M_j)$:

$$M_s = M_p + M_j \tag{2}$$

By applying fast Fourier transform (FFT) to $M_s$ and assuming uniform amplitude for the excitation torque of each cylinder, the harmonic torque of different engine cylinders can be obtained [3].

## 3. Engine Modifications and Mechanism Design

### 3.1. Engine Modifications

Cylinder balancing is a well-established technique for reducing torsional vibrations arising from unbalanced cylinders in common-rail engines and attenuating transmitted disturbances to the vehicle's passengers. In this approach, cylinder fuel injectors are controlled electrically to adjust the injection durations and timings, by stabilizing the dynamic loads and providing a uniform mechanical strain distribution in engine components, such as the crankshaft through time- and frequency-domain analyses [4]. Time-domain balancing method is more demanding in the automotive industry [5,6] as it only requires measurement of the speed at the flywheel and taking the numerical integration of the estimate torques applied to it. On the other hand, in frequency-domain methods, the ultimate goal is to directly reduce the harmful vibration frequency components through computationally -efficient calculations, such as DFT-based control of cylinder injections [7] and artificial neural networks for pattern-recognition of crankshaft angular speed waveforms [8].

Saxén et al. [9] applied time- and frequency-domain power balancing algorithms to adjust the fuel-injection and reduce power unbalance of a 7-cylinder diesel engine-generator, with a flexible crankshaft, through speed measurements at 750 rpm. The proposed approach was able to balance the estimated powers via reconstructing the produced torque at the flywheel in four iterations and tuning the fuel injection durations without affecting the total power of the engine. The outcome was an 82% reduction in undesirable torsional vibrations induced by power unbalances. Figure 1 illustrates an excellent consistency between the simulation and full-scale test results on the engine, using their power balancing algorithm.

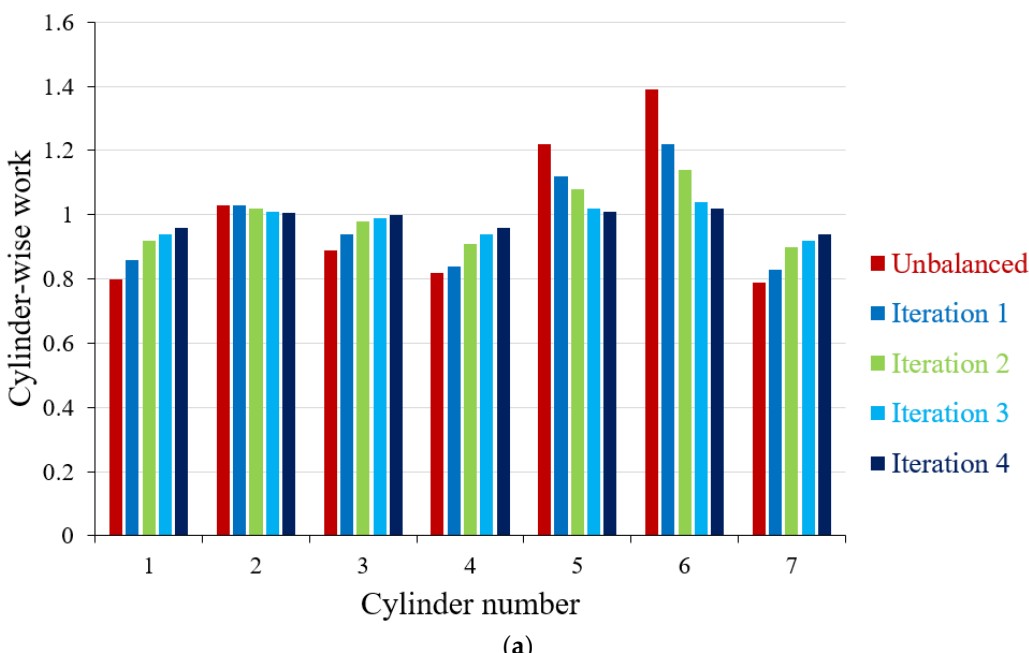

(a)

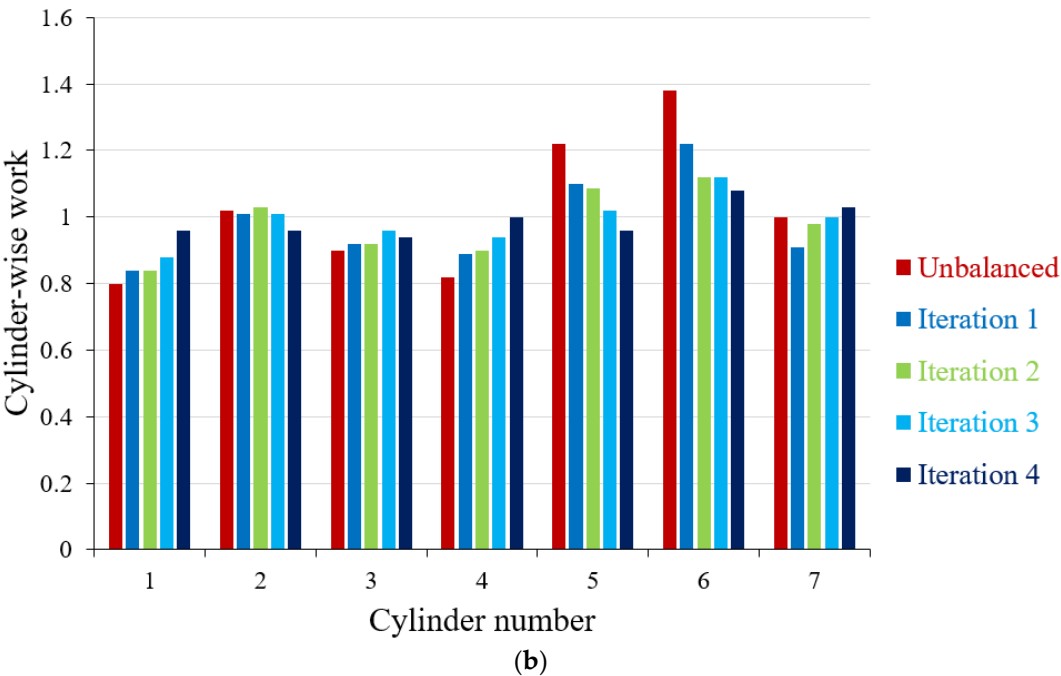

**Figure 1.** Normalized cylinder-wise torque integrals in power balancing results: (**a**) simulations; (**b**) actual test on a medium-speed 7-cylinder diesel engine [9].

Engine mounting system plays an inevitable role in isolating the driver and passenger from the noise, vibration, and harshness (NVH), produced by power-intensive engines in modern lightweight vehicles [10, 11]. An ideal engine mount should feature (i) high stiffness and excellent damping capability to isolate the frequency-dependent disturbances excited by powertrains and road vibrations; (ii) support static powertrain weight and dynamic engine torque; and (iii) hinder engine bounce from shock excitation [12,13]. Therefore, the performance and reliability of these systems require further improvements for a better NVH refinement.

Engine mounting transmission ratio (EMTR), or transmissibility, which is defined as the ratio of the vibration generated at the engine to the one that is transmitted through the mounting system to the body of the vehicle, is a useful metric for expressing isolation efficiency of the engine mounts at a specific frequency or speed [14]. While many statistical tools have been developed to detect and analyze different physical and natural phenomena [15–18], a well-established measure in vibration diagnosis and analysis is the power content (root mean square or RMS) value of the measured vibration signal [19–21].

Mohamed et al. [14] performed time- and frequency-domain analyses on a 4-cylinder engine, running at 800–4000 rpm, to evaluate the vibration isolation and transmissibility of three rubber engine mountings (REM), including cylindrical (A), cone-and-shear (B), and center-bonded mounts (C). They concluded that a cone-and-shear system offers the minimum RMS of vibration accelerations (Figure 2) and optimum transmission ratio when it is mounted on the engine and chassis (Figure 3).

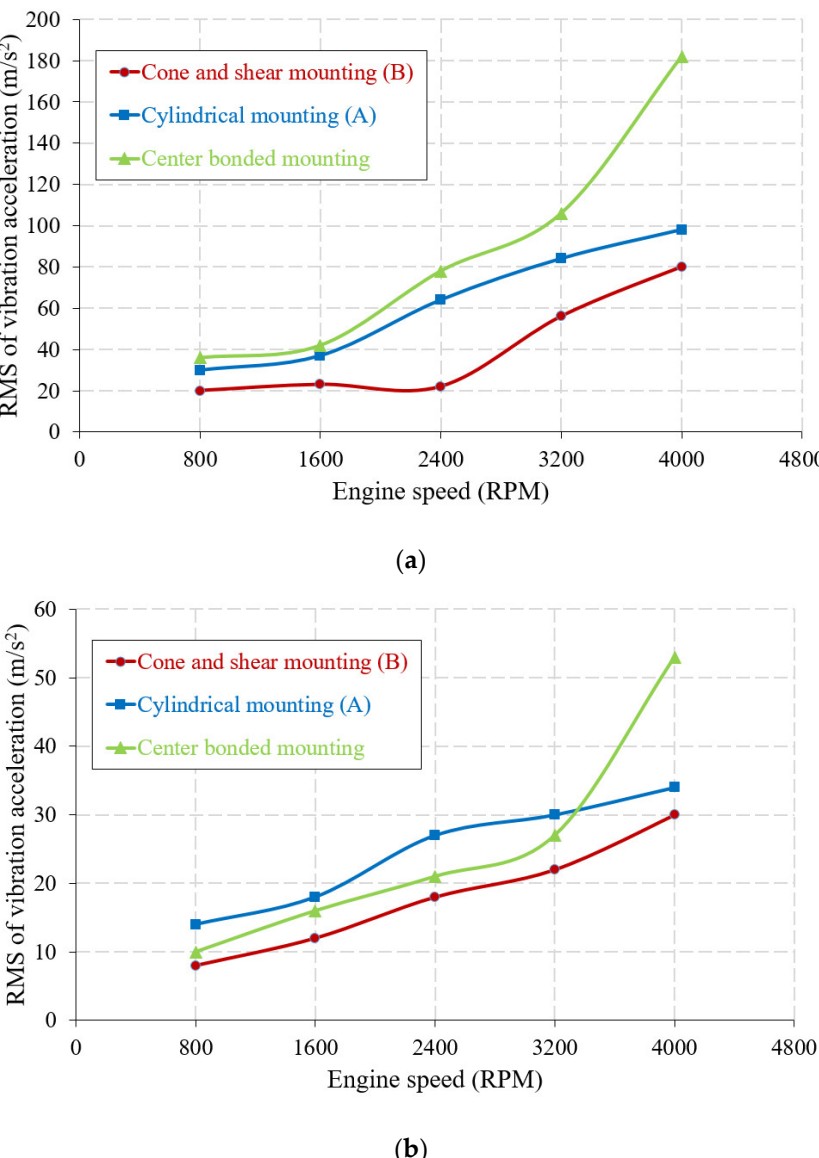

(**a**)

(**b**)

**Figure 2.** RMS of vibration accelerations of (**a**) engine; (**b**) chassis [14].

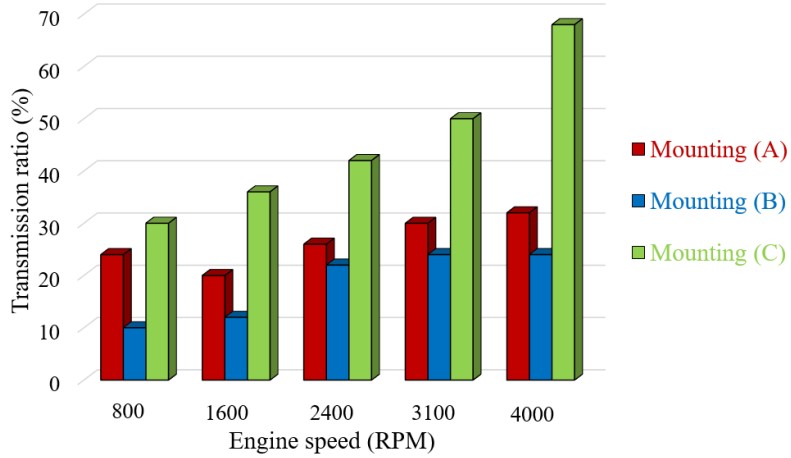

**Figure 3.** The transmission ratio of each mounting system at different engine speeds [14]. The rubber cone-and-shear mount offered the highest vibration isolation at all speeds.

A further enhancement was achieved by adding a concentrated mass (steel block) of ~1.5 kg, to the location of the rubber cone-and-shear engine mount, on the chassis. In that way, the spike acceleration of the chassis dropped more than two times, when the engine was operating at 1600 rpm. This modification also improved the transmission ratio of the mount, from 14% at 800 rpm, up to 19% at 4000 rpm (Figure 4).

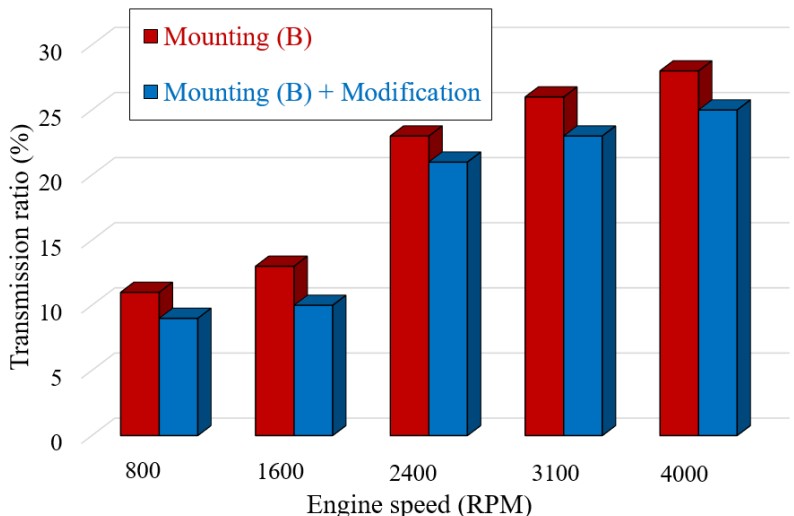

**Figure 4.** Improvement in transmission ratio after applying the concentrated mass at all speeds [14].

In their experimental study, Manhertz and Antal [22] studied the effect of air–fuel ratio and ER ($\lambda$) on vibration components of a gas-powered ICE equipped with an asynchronous generator. Measurements were carried out at a constant speed (1500 rpm) to maintain the frequency of the components unchanged. A short-time Fourier transform (STFT) analysis was used to determine the spectral components. During the experiments, four time signals with varying oxygen ($O_2$) contents of the exhaust gas were recorded to measure the maximum and minimum amplitudes at different ERs (Table 1).

**Table 1.** The maximum and minimum amplitudes of the four-measurement setup as a function of the air-fuel equivalence ratio [23].

| $\lambda$ (-) | $O_2$ (%) | $A_{max}$ (s) | $A_{min}$ (s) |
|---|---|---|---|
| 1.06 | 1.3 | 46.57 | −54.65 |
| 1.24 | 3.9 | 36.81 | −38.31 |
| 1.51 | 7.6 | 31.51 | −32.49 |
| ∞ | 0.0 | 29.38 | −28.97 |

Here, ∞ means that no combustion occurred in the chamber and the engine was actuated only by the asynchronous machine. As shown in Figure 5, the amplitudes tended to drop with increasing the ER function (% $O_2$). In other words, the vibration components decreased by degradation of the air–fuel mixture, i.e., the step-by-step garbling of the perfection of the combustion.

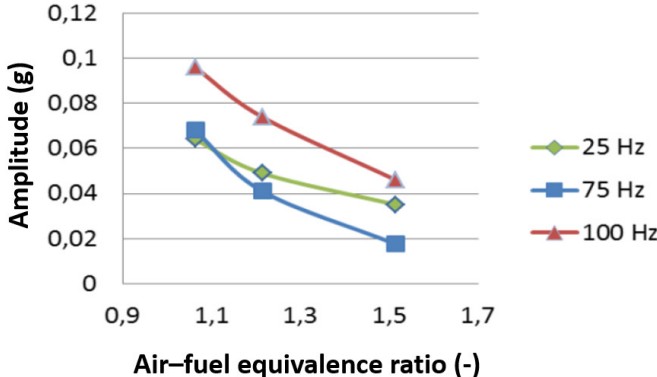

**Figure 5.** Amplitude changes of some frequency component as a function of ER [23]. Figure is reprinted with permission from publisher.

Borg et al. [23] investigated mitigation of noise and vibration in a direct injection gasoline (DIG) engine by optimizing the geometry of its high-pressure fuel pump, without affecting the fuel consumption and emissions of the engine. Figure 6a illustrates schematic of a typical DIG, composed of different components. Fuel system, the heart of the DIG, contains a high-pressure fuel pump, fuel pipes, and injectors. It is considered as the main source of noise generation in the DIG engines. Since the pump and injectors are typically solenoid-actuated, the vibration can be reduced or diminished if accurate design and control modifications are applied to the system. For the former, vibration damping could be improved by applying changes to the fuel rail/cylinder head and injectors/cylinder head mountings. Hence, those researchers sandwiched a rubber isolator between the fuel rail clips and cylinder head, to isolate the system noise and eliminate the vibration transmission path to the cylinder head (Figure 6b). Furthermore, the cylinder head vibrations were suppressed by suspending the injectors from the fuel rail, providing no mechanical contact between the components. Also, they were able to attenuate the compression (at outlet) and ticking (at inlet) noises of the solenoid (up to 5 dB), using soft-landing and pulse-skipping control techniques, respectively.

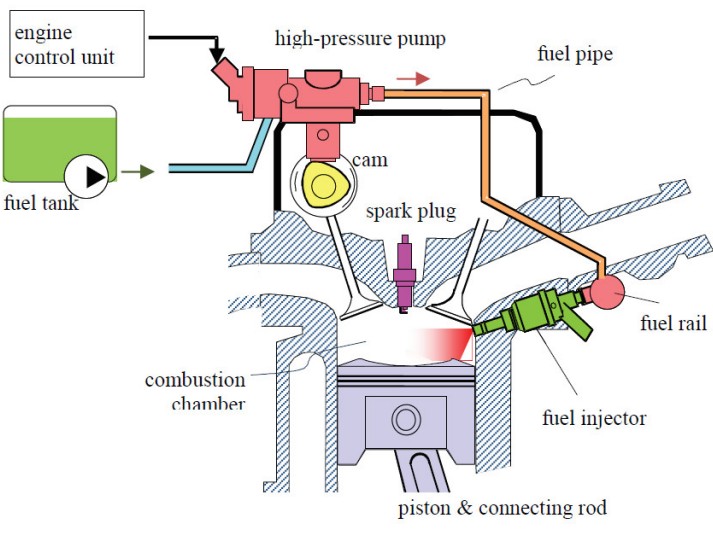

(a)

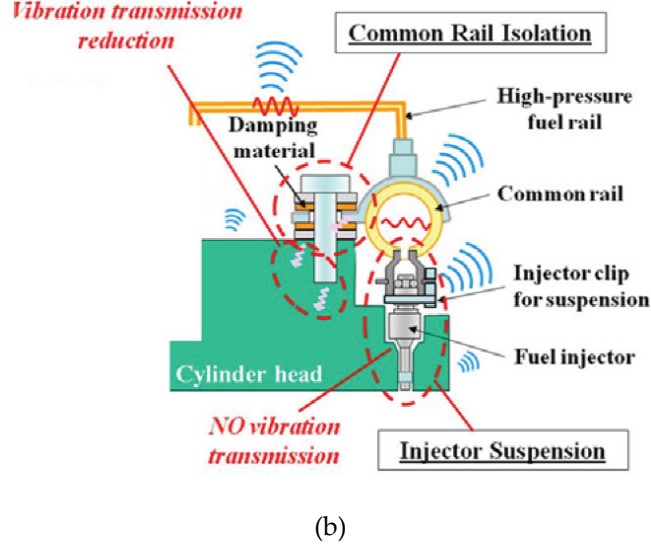

(b)

**Figure 6.** Schematics of (**a**) gasoline direct injection fuel system; (**b**) rail isolation and injector suspension [23]. Figures are reprinted with permission from publisher.

Zavos et al. [24] experimentally and theoretically predicted the generated noise and vibration in the piston-rack assembly of a single-cylinder motorbike engine. The friction of the piston-ring pack system was directly measured along the cylinder bore, using a foil strain gauge, while the noise level was recorded at the thrust side, using a microphone, to estimate the tribological conditions of the engine such as piston slaps. Their findings highlighted that, with a lower piston speed and higher cylinder pressure, the friction force is pushed towards the boundary/mixed regime, exposing the engine to a higher level of noise, which is generated in the transition between the compression and power strokes.

In another study, Dolatabadi et al. [25] developed a novel analytical–numerical method to investigate the fundamental role of piston slap in NVH of ICEs. They comprehensively analyzed the contribution of elastohydrodynamic lubrication (EHL) and piston transient dynamics in the detection of slap noise events. The impact force and squeeze film velocity were estimated based on the conjunctional pressures and the oil film thickness variations. Furthermore, the piston secondary motion was correlated to the surface vibrations of the engine block to calculate the noise levels. Their proposed method enabled an accurate prediction of time and frequencies of the piston slap events during the engine cycle, and simple detection of noise amplitudes with a monotonic increase in the engine speed.

### 3.2. Mechanism Design

Kosenok et al. [26] analyzed the dynamics of the slider-crank mechanism of a two-cylinder ICE analytically, using vector modeling. Figure 7 demonstrates schematic of the studied V-shaped ICE in this work. Their research aimed at finding optimum geometrical parameters, including the angle between the axis of the engine cylinders $(a_{cyl})$ and the angle between the crank and cylinders $(a_{cr})$, for which balancing moments and vibrational loads on the motor were the least.

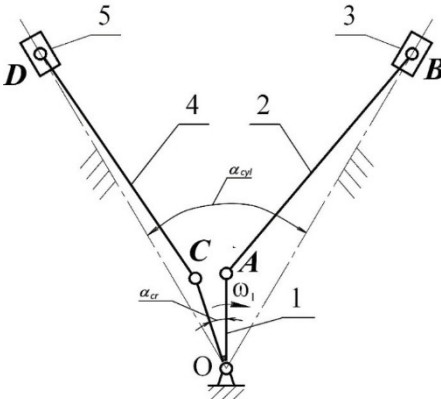

**Figure 7.** (**a**) Kinematic scheme of V-shaped ICE: 1—crankshaft, 2 and 4—couplers, 3 and 5—pistons [26]. Figure is reprinted with permission from publisher.

Figure 8 depicts the amplitude of balancing moments of inertia ($\Delta J_R$) and coefficient of mechanical losses ($K_{ML}$) for $0° \leq a_{cyl} \leq 360°$. Each curve exhibits the dynamic behavior with respect to the angle between the cranks ($a_{cr}$), ranging from 0° to 180°. As can be seen in Figure 8a, the maximum amplitudes of the moment of inertia occurred at $a_{cr} = a_{cyl} + 0°$ or $\pm 180°$. This horizontal layout could be used for applications where the engine is expected to endure high mechanical losses and stresses. On the other hand, they showed that by defining the optimum angle of the crankshaft as $a_{cr} = a_{cyl} \pm 90°$, cylinders generate less longitudinal (horizontal) vibrations. This finding is of particular interest in aviation applications where high-speed ICEs, such as aircraft, are required to attenuate the vibrations aroused by high dynamic loads.

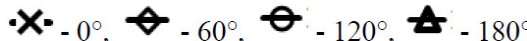

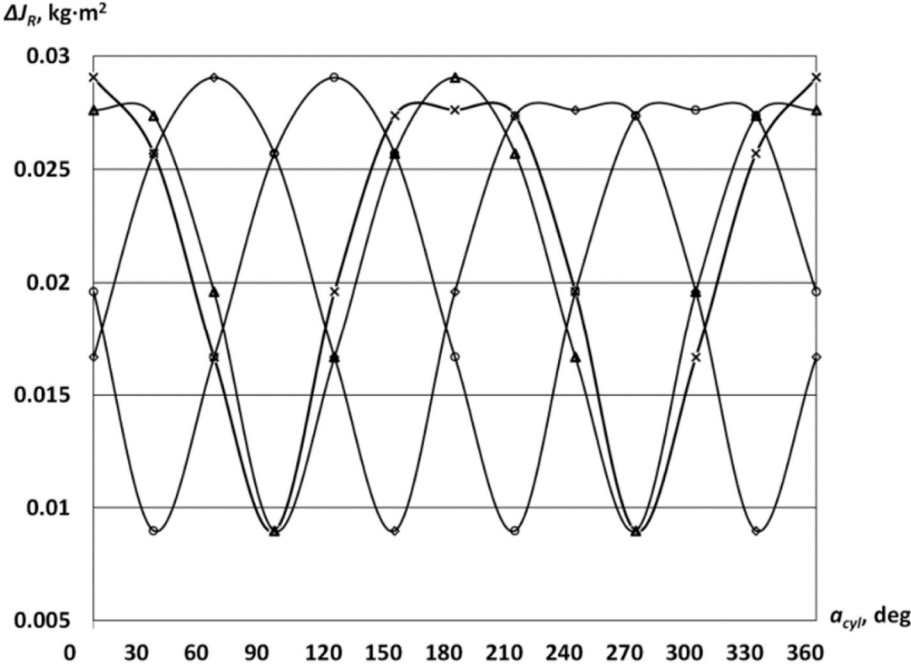

(**a**)

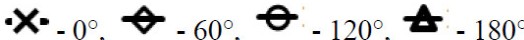

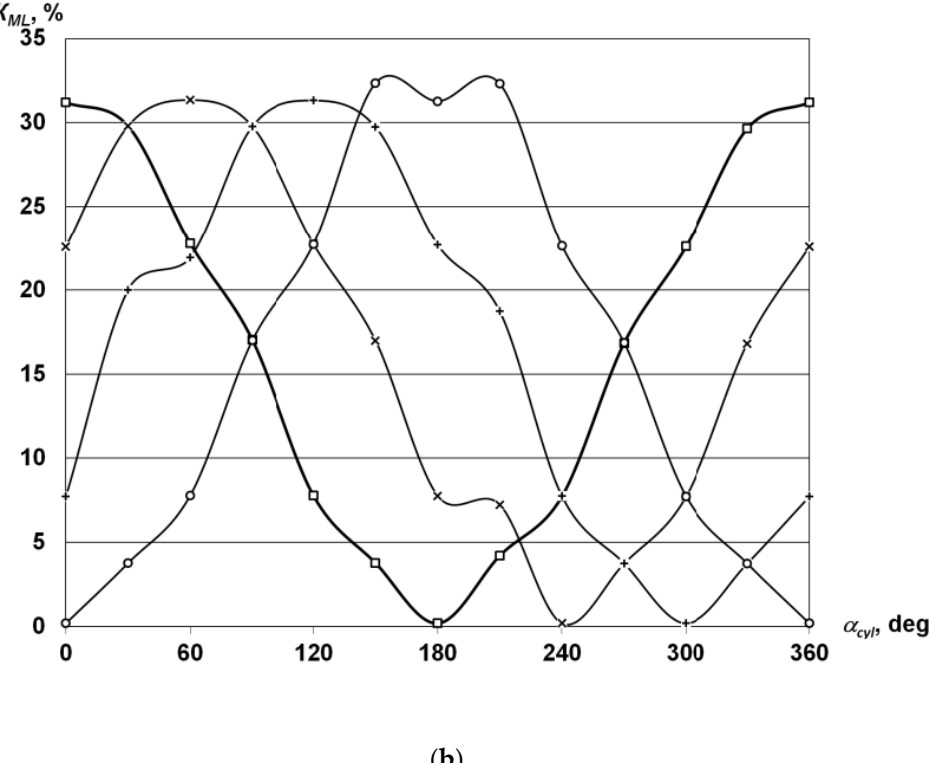

(b)

**Figure 8.** (**a**) Amplitudes of the balancing moment of inertia and (**b**) the coefficient of mechanical losses, with respect to $\alpha_{cyl}$, for $0° \leq a_{cr} \leq 180°$ [26]. Figures are reprinted with permission from publisher.

Pfabe and Woernle [27] synthesized a kinematically driven flywheel (KDF) to reduce the torsional crankshaft vibrations of an ICE through compensation of fluctuating engine torques, directly at the location where they arise. In their model, the flywheel was coupled to the crankshaft via a transmitting mechanism (Figure 9a). The proposed mechanism enabled a non-uniform conversion of the crankshaft rotation angle into the flywheel rotation angle to compensate the inertial flywheel torques ($\theta_{FW}\ddot{\psi}$) by at least one harmonic of the fluctuating engine torque (Figure 9b). Since amplitude and phase angle of the excitation torque changed with the engine speed ($\dot{\varphi}$) and applied load, they prescribed a continuous adjustment of the compensated torque, $M_{FW}(\varphi)$, with respect to the engine state. Also, due to the high amplitudes of the compensated torque generated by the non-uniform flywheel inertia, a direct force flow through the mechanism must be ensured.

To meet the requirements mentioned above and expand the compatibility of their transmitting mechanism to ICEs with single/multiple-cylinders, those researchers also developed a geared double-crank mechanism ($\overline{ABCD}$) with cycloidal-crank input ($\overline{BC}$) to couple the flywheel ($\overline{AD}$) to the crankshaft ($\overline{AB}$) (see Figure 10). The amplitude of the non-uniformity of the kinematic transfer function, $|\psi(\varphi)|$, was adjusted by the eccentricity ($e$) of the input crank. The topology of this mechanism enabled generating high relative accelerations, between the crankshaft and flywheel, and triggered high amplitude compensating torques, at a four-stroke three-cylinder inline diesel engine, under full load at 2000 rpm.

Figure 11a represents that the correlation between measured and simulated $M_{FW}(\varphi)$ had a good agreement at 600 rpm and 4 mm eccentricity. The close agreement between the compensated torque, measured from forces in the connecting rod and values measured directly from the experiments, assured the consistency of the measurements (Figure 11b).

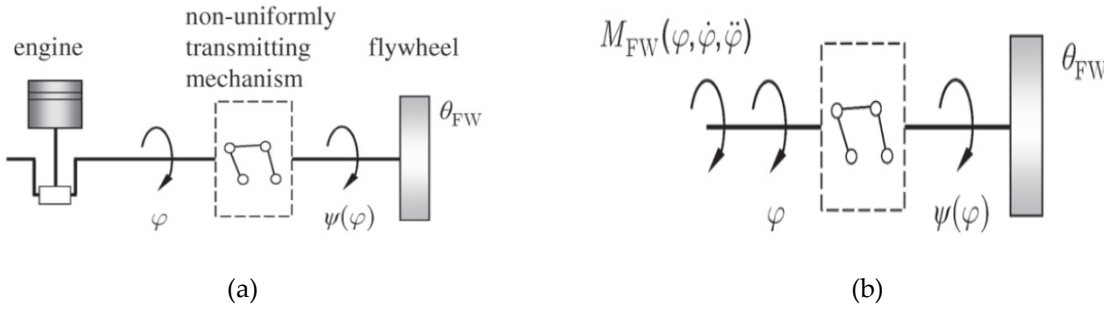

**Figure 9.** Kinematically driven flywheel (KDF) of a combustion engine: (**a**) schematic of the arrangement; (**b**) free-cut compensation system [27]. Figures are reprinted with permission from publisher.

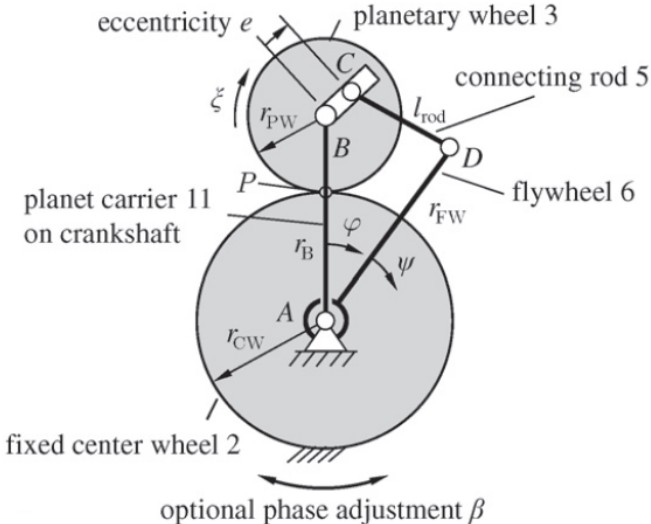

**Figure 10.** Mechanism topology of the proposed KDF [27]. Figure is reprinted with permission from publisher.

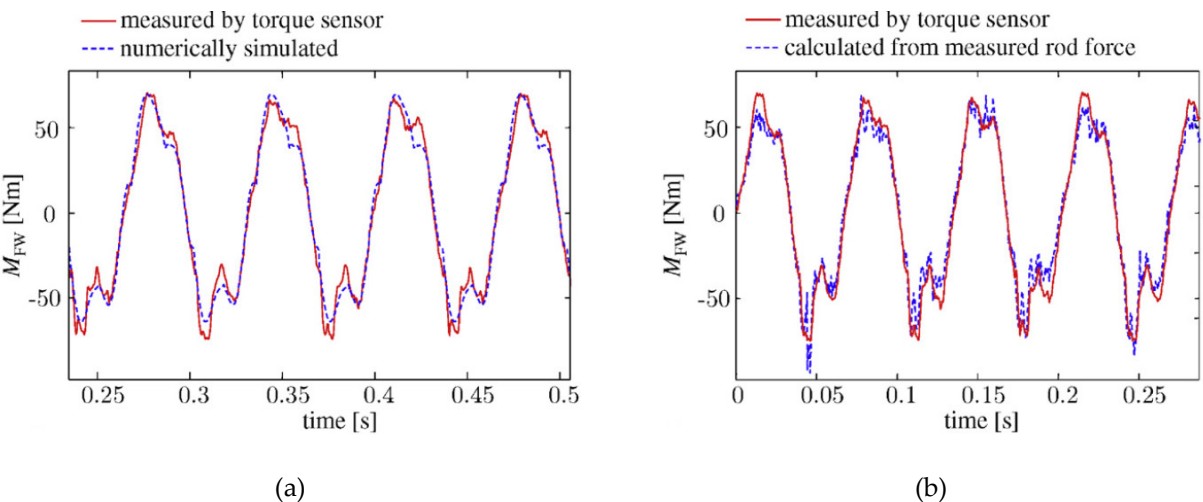

**Figure 11.** (**a**) Numerically simulated non-uniform part of the engine torque $\Delta M_{Eng}(\varphi)$ and compensation torque $M_{FW}(\varphi)$ of the synthesized KDF. (**b**) Flywheel angular speed $\dot{\varphi}_{ideal}$ for complete compensation of $\Delta M_{Eng}(\varphi)$ and flywheel angular velocity $\dot{\psi}(\varphi)$ of the synthesized KDF [27]. Figures are reprinted with permission from publisher.

In a computer-aided engineering-based (CAE-based) study conducted by Zhang et al. [28], finite element analysis (FEA) and multi-body analysis (MBA) tools were implemented for modeling the dynamics of acoustical components of a six-cylinder 6160 in-line diesel engine, to understand the interaction between excitation mechanisms and noise transmission in the system (see Figure 12).

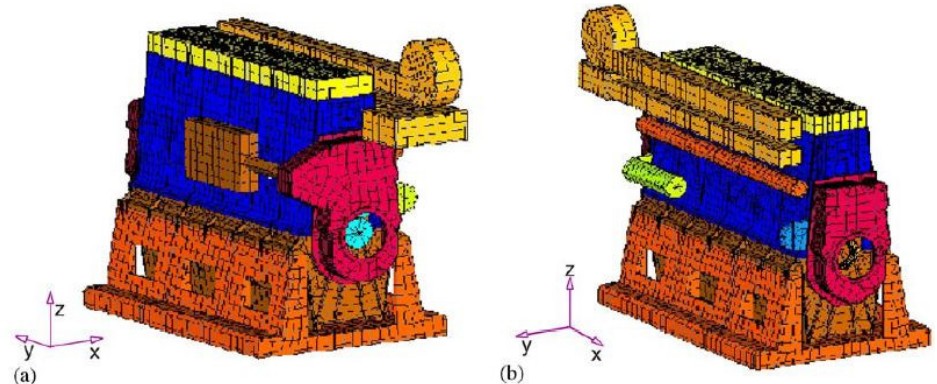

**Figure 12.** Multi-body analysis (MBA) model of a 6160 diesel engine. (**a**) Left and (**b**) right part of the engine [28]. Figures are reprinted with permission from publisher.

The research was aimed at applying structural modifications to reduce noise and optimize vibrations at the engine, while operating at 400–1000 rpm, under full load, and in low-frequency range of <1 kHz. Whereas the FEA was conducted to analyze the vibrational behavior of the housings and crank train/valve drive of the engine, the MBA was utilized to model the dynamic behavior of rotating components (e.g., transmission shafts, gears, timing drive, etc.) and simulate vibroacoustic responses of the whole system. Forced vibration calculations were performed in the time-domain, under various speed and load conditions. The simulations were carried out in a direction perpendicular to the engine surface, to obtain rpm spectral maps at different vibration levels. Results discerned that most of the engine surface vibrations were induced at a range close to the natural frequencies of the crankshaft, and at a speed between 600 to 1000 rpm. To minimize the vibration transferred through the internal paths and enhance the structural stiffness of the engine, two major design modifications were proposed: (i) increasing the local wall thickness of the crankcase, front gear cover, and back flywheel cover; (ii) adding ribs (extra mass of 14.85%) to the drive shaft of the engine which added extra supports for the drive shaft bearing bracket and more stiffness to the oil sump. Figure 13 demonstrates that the modifications reduced the vibration intensity of the original design for the modes between 0 to 1 kHz.

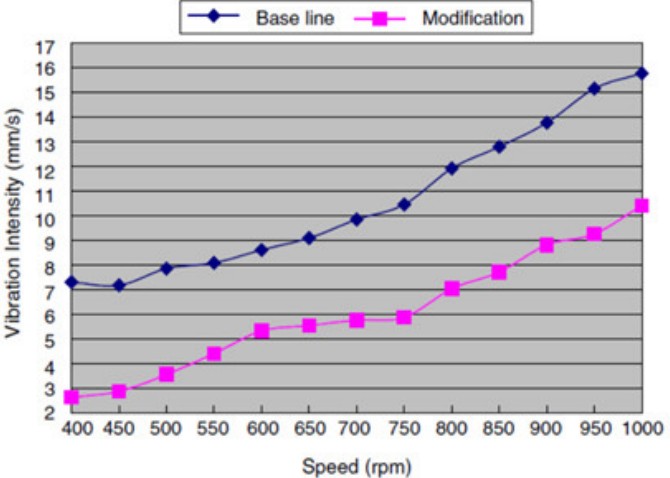

**Figure 13.** Modal density comparison of the original vs. modified engine designs [28]. Figure is reprinted with permission from publisher.

Meng and Li [2] investigated the impact of the geometric and operation parameters of a piston–crankshaft system on the dynamic performance of a single-cylinder four-stroke ICE (Figure 14a). In their proposed analytical model, the coupling between the components and oil film forces of the lubricated components were simultaneously incorporated into the dynamic response of the piston pack. For developing the dynamic equations of the system, an average flow model (Reynolds equation) [29,30] was adopted, based on classical lubrication theory, to take into account the effect of the oil film forces of the piston rings and piston skirt. The $x$ and $y$ components of the oil force, $F_o$, exerted by the main bearing were calculated as

$$F_{ox} = \int_0^L \int_0^{2\pi} pR_2 \sin \phi \, d\phi \, dz, \tag{3}$$

$$F_{oy} = \int_0^L \int_0^{2\pi} pR_2 \cos \phi \, d\phi \, dz, \tag{4}$$

where z is the width direction of the main journal, $\phi$ is the circumferential angle measure from the negative $x$-axis (Figure 14b), and $L$ is the width of the main bearing with inner radius of $R_2$. Moreover, the hydrodynamic lubrication behavior of the lubricant was simulated at the piston pack and cylinder liner interface, as well as the main journal and main bearing interface. Once the oil forces for the piston and main journal were obtained, the dynamic equations of the piston–crank system were developed and validated numerically [31]. The vibration differential equations for the main journal were expressed as

$$m\ddot{X} + q\dot{X} + kX + F_{ox} - F_{01x} + F_{ex} = 0, \tag{5}$$

$$m\ddot{Y} + q\dot{Y} + kY + F_{oy} - F_{01y} + F_{ey} = 0, \tag{6}$$

where $m$ and $q$ are the mass and damping coefficient of the main journal, respectively. $X$ and $Y$ are the displacements of the main journal, and $F_{01x}$ and $F_{01y}$ are the reactions applied by the main journal on an initiated crack. The components of the unbalanced force $F_e$ were computed as

$$F_{ex} = m_1 e_1 \omega^2 \sin \omega t, \tag{7}$$

$$F_{ey} = m_1 e_1 \omega^2 \cos \omega t, \tag{8}$$

where $m_1 e_1$ is the torque exerted by the eccentricity of the unbalanced mass of the crankshaft, with respect to its original center. The $X$ and $Y$ displacements of the main journal were then determined from the dynamic equations, at varying connecting-rod length to piston diameter ($L_c/D_s$) ratios. Although the ratios exhibited minimal effect on orbit of the main journal center and $Y$ displacements, they significantly affected the $X$ displacements. With increasing $L_c/D_s$, the vibration amplitude of the main journal was increased along the $x$-direction, while it was dropped with negligible effect on the $Y$ displacement.

Damping coefficient, $q$, is another parameter that influences the orbit and vibration amplitude of the main journal center. The researchers also investigated the effect of different damping coefficients on the relative difference in vibration displacements and eccentricity ratios of the main journal, in the 1500–3000 N s/m speed range. While the orbits were relatively stabilized (i.e., round loop shape), with increasing the $q$, a further increase made the relative differences in vibration amplitudes larger and, thus, degraded the hydrodynamic action (vibration attenuation) of the lubricant between the main journal and main bearing.

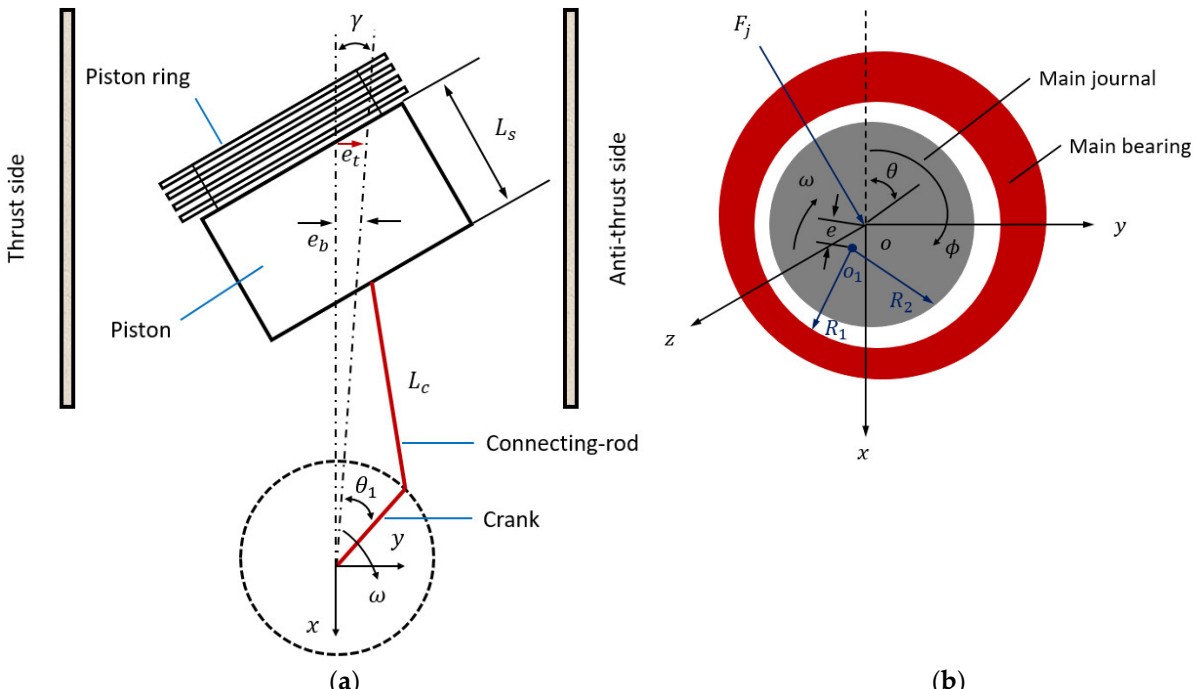

**Figure 14.** Schematic of the piston–crankshaft system: (**a**) piston-cylinder; (**b**) main bearing–journal system [2].

To highlight the contribution of engine lubricating oil quality and condition, Albarbar et al. [32] remotely measured the surface-borne and airborne acoustic signals caused by piston slap, and studied the effects of load, engine speed, and oil temperature. They reported that a reduction in the oil level and/or the use of poor quality oil can cause an increase in structure-borne acoustic and airborne acoustic signals of certain frequency bands.

Shadloo et al. [33] designed a crank and slider mechanism to enhance the combustion performance of a four-stroke spark ignition (SI) engine. By lowering the ratio of the engine's connecting-rod length to crank radius ($l/R$), the isentropic process in the power stroke was desirably resembled, leading to a reduction in the maximal gas temperature after the first spark and increase in the compression ratio. These two factors improved the thermodynamic (heat) efficiency of the engine and increased the linear speed of the piston within the range of combustion. Particularly, with increasing the crank angular velocity, higher friction efficiency and less engine noise were achieved, through removing the piston lateral pressure and providing no contact condition between the piston and cylinder barrel. These improvements in the engine design and thermodynamic performances were accompanied by lower NOx pollutions and a longer life service of the engine. However, there were some drawbacks associated with their proposed mechanism, such as increasing the number of the engine components, complexity of the mechanism, and barriers in assembling the engine parts.

Lin et al. [34] synthesized a novel balancing cam mechanism for a single-piston engine to passively compensate resistive torque fluctuations of an engine camshaft at all positions and speeds. The camshaft was comprised of inlet and exhausted cams, each separately connected to a set of rocker, valve, and spring (see Figure 15). Three major assumptions were made in analysis of the camshaft torque: (i) the inlet and exhaust springs had identical geometry, stiffness ($k$), and preload ($P$); (ii) the engine was well-lubricated and frictionless; and (iii) the rollers were treated as two massless two-force members, with the contact force applying to their centers.

Considering the camshaft rotates at a constant speed, the force and moment equations for inlet and exhaust components of the camshaft were derived using the free-body diagram (FBD) of each component:

$$|\mathbf{F}_{sX}| = (k\delta_X + P), \text{ (spring)} \tag{9}$$

$$\mathbf{F}_{sX} + \mathbf{F}_{nX} = m_{vX}\ddot{\delta}_X \mathbf{e}_X, \text{ (spring)} \tag{10}$$

$$\mathbf{r}_{aX} \times \mathbf{F}_{cX} + \mathbf{r}_{bX} \times (-\mathbf{F}_{nX}) = I_X \ddot{\phi}_X \mathbf{k}, \text{ (rocker)} \tag{11}$$

$$T_{cX}\mathbf{k} + \mathbf{r}_{cX} \times (-\mathbf{F}_{cX}) = 0, \text{ (cam)} \tag{12}$$

$$T = T_{ci} + T_{ce}, \text{ (camshaft)} \tag{13}$$

where subscript $X$ could be replaced with either inlet or exhaust notations in Figure 16 (subscripts $i$ and $e$). The description of each notation is represented in Table 2.

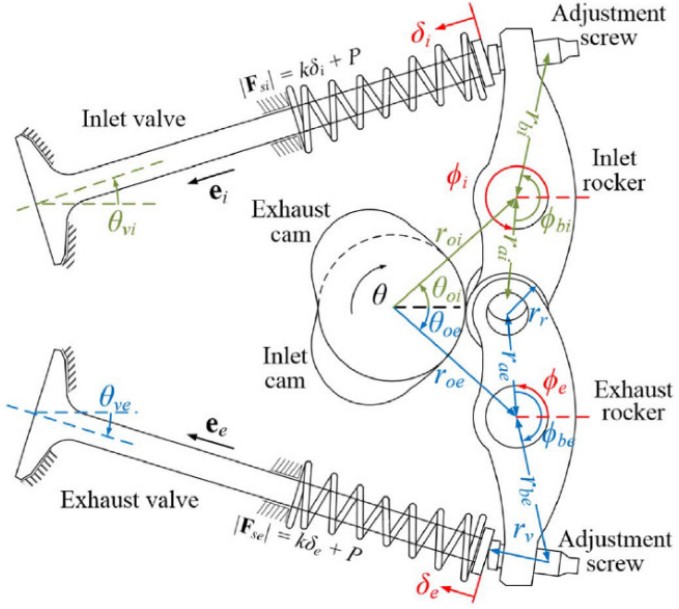

**Figure 15.** Schematic of the balancing mechanism [34]. Figure is reprinted with permission from publisher.

**Table 2.** Notations used in derivation of the force and moment equations for the camshaft analysis.

| Notation | Description | Notation | Description |
|---|---|---|---|
| $m_{vX}$ | Mass of inlet/exhaust valve | $\mathbf{F}_{sX}$ | Inlet/exhaust spring force |
| $\mathbf{F}_{nX}$ | Force from adjustment screw to inlet/exhaust valve | $\mathbf{F}_{cX}$ | Force from inlet/exhaust cam to rocker |
| $\mathbf{r}_{aX}$ | Position vector from rocker pivot to roller center | $\mathbf{r}_{bX}$ | Position vector from rocker pivot to adjustment screw |
| $I_X$ | Moment of inertia of inlet/exhaust rocker | $\mathbf{r}_{cX}$ | Position vector from inlet/exhaust cam axis to roller center |
| $T_{cX}$ | Required inlet/exhaust torque on camshaft to balance with force $\mathbf{F}_{cX}$ | $T$ | Total camshaft torque |
| $\delta_X$ | Inlet/exhaust valve linear displacement | $\phi_X$ | Inlet/exhaust rocker angular displacement |
| $\mathbf{e}_X$ | Unit vector along inlet/exhaust valve | $\mathbf{k}$ | Unit vector along rotation vector |



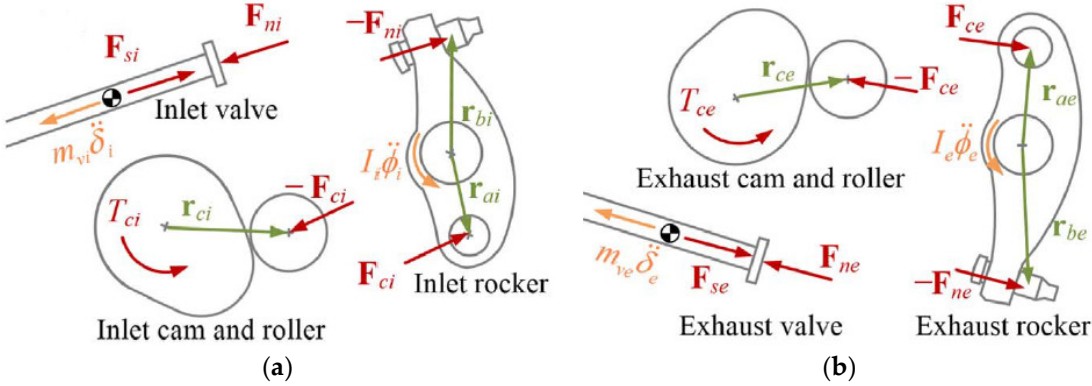

**Figure 16.** Free-body diagram (FBD) of the valve, rocker, cam, and roller of the engine camshaft at: (**a**) inlet; (**b**) exhaust [34]. Figures are reprinted with permission from publisher.

By analyzing the inlet and exhaust valve displacements for one full revolution of the cam, they showed that large spring force variations were responsible for the fluctuation of the resistive camshaft torque. These dynamic speed fluctuations were recognized as the main vibration source of the camshaft component. To resolve this issue, a balancing mechanism was proposed, consisting of a cam, rocker, spring, and spring guide, similar to the valve structure (see Figure 17). From the FBD of the balancing mechanism and applying the same equations as Equation (7) to (11), the equation of the resistive torque generated by this mechanism was developed. The main purpose of this balancing mechanism was to cancel the resistive torque generated by the valve mechanism, at any rotation angle of $\theta$, and gives a zero-output torque. To validate and explore the capability of the proposed mechanism, numerical simulations were conducted, and results showed that this methodology could be extended to multi-piston engines. In resemblance to methods of using flywheel, this mechanism was able to operate at all speeds, with no input power, and thus zero energy loss.

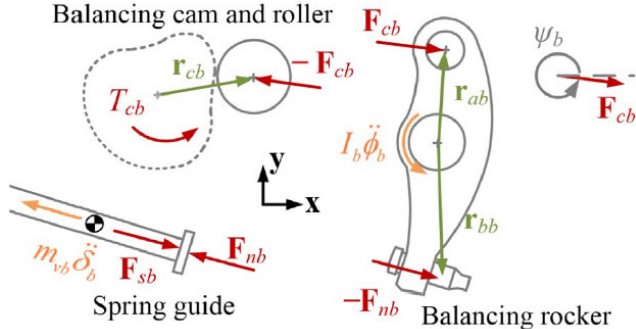

**Figure 17.** FBD of the proposed balancing mechanism [34]. The mechanism was able to cancel the resistive torque generated by the valve mechanism. Figure is reprinted with permission from publisher.

## 4. Biodiesel Fuels

Biodiesel is an environmentally friendly, renewable fuel which can be used in any compression ignition engine to substitute diesel fuel. Chemically, biodiesel contains a mixture of methyl esters with long-chain fatty acids, and is typically derived from nontoxic biological resources, such as vegetable oils [35–39], animal fats [40–46], or even cooking oils [47–50]. While many recent works analyzed the effects of biodiesel on the injection process, spray formation, and combustion [51–56], this section will emphasize the potential of different biofuel blends in reducing the noise and chemical pollutions of ICEs.

Taghizadeh-Alisaraei et al. [57] investigated the effect of different diesel–biodiesel blends on performance and vibration characteristic of a 6-cylinder diesel engine running at 1000–2200 rpm. Nine fuel blends were studied: B0D100 (pure petrodiesel), B5D95 (5% biodiesel + 95% diesel), B10D90, B15D85, B20D80, B30D70, B40D60, B50D50, and B100D0 (pure biodiesel). The biodiesel fuels were prepared from vegetable oil, animal fat, and waste oil, according to ASTM D6751-09 standard. The combination of the fuel blends and engine speeds generated several experiments. Time- and frequency-domain vibration analyses were conducted at various engine speeds. The RMS of acceleration for each experiment was calculated:

$$a_{RMS} = \sqrt{\frac{\sum_{k=1}^{N} a_k^2}{N}}, \tag{14}$$

where $a_{RMS}$ is the root mean square of acceleration signal $a_k$, $k$ is the acceleration value in the time-domain signal, and $N$ is the total number of acceleration data ($N = 80{,}000$), obtained for a duration of 1.6 s. For a multiple comparison, differences between the mean values of the experiments were compared, using the balanced 2-factorial analysis of variance (ANOVA) and Duncan's multiple range test (DMRT). Also, the total acceleration ($a_t$) value was calculated from vertical ($z$), lateral ($y$), and longitudinal ($x$) components of the acceleration signal:

$$a_t = \sqrt{a_x^2 + a_y^2 + a_z^2}. \tag{15}$$

The combustion of different fuel blends does not follow a systematic trend. Although an increase in the fraction of biodiesel in the fuel blend is expected to enhance fuel viscosity and reduce the engine vibration due to incomplete combustion (low heating value and power) [52], the noise and engine vibration depend on many other factors, such as lubrication properties, cetane number (CN), flash point, thermal/physical properties, chemical/molecular structure of the fuel blends, oxygen level during combustion, and injection/spraying of the fuel. The cetane number and viscosity increases with biodiesel addition to the fuel blend, while the cylinder peak pressure drops [58,59]. As illustrated in Figure 18, before the service engine, B40 and B20 biofuel blends had the lowest mean $a_t$ values. These results, together with analysis of the vibration signal, suggested that among the other biodiesel blends, B40 and B20 had a minimal effect on engine knocking, due to production of small and smooth gas pressure changes (fluctuations) within the cylinder chamber upon, and during, the combustion process.

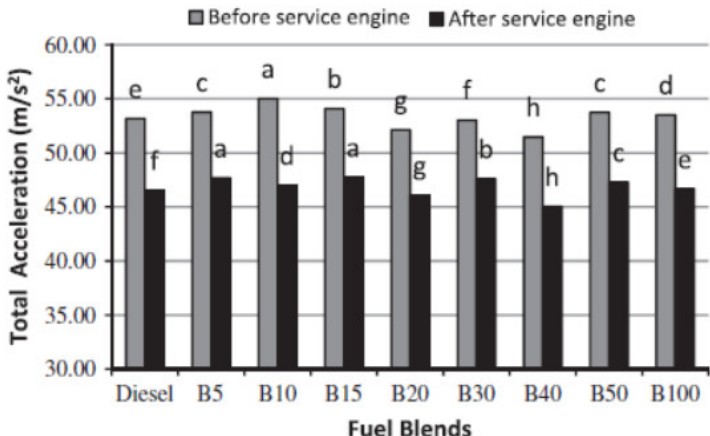

**Figure 18.** Total vibration acceleration ($a_t$) of different diesel–biodiesel fuel blends, before and after the service engine [57]. Figure is reprinted with permission from publisher.

A diesel engine process includes fuel injection, spraying, mixing, ignition, and combustion. Among these stages, the fuel injection and fuel spray are highly affected by the fuel quality. The fuel

spray strongly depends on the injection delay, injection pressure, dynamic injection timing (DIT), injection duration, and injection nozzle configuration [58]. Compared to diesel, biodiesel fuel offers better physicochemical properties (i.e., density, viscosity, distillation, bulk modulus, surface tension, and others), to control the injection process [52,58].

In diesel engines, knocking mainly occurs due to (i) high ignition delay (i.e., long time gap between the fuel injection and combustion); and (ii) sudden impacts exerted on the cylinder head during opening–closure, imposed by large valve clearance. In a related effort, Taghizadeh-Alisaraei et al. [60] studied the correlation between biodiesel concentration and combustion, vibration, and knocking, in single- and six-cylinder diesel engines. Six biodiesel–diesel fuel blends were prepared using waste cooking oil, according to ASTM D6751-09 standard: D100 (pure petrodiesel), B20 (20% biodiesel + 80% diesel), B40, B60, B80, and B100 (pure biodiesel). The contents of saturated and unsaturated fatty acids were measured through gas chromatography (GC), and the cetane number of the fuel blends was determined using ASTM D613. The engine vibration signals were measured in vertical ($z$), lateral ($y$), and longitudinal ($x$) directions, using three accelometers. Short-time Fourier transform (STFT) and Morlet scalogram (MSC) distribution algorithms were exploited for knocking and fault detection [61–67]. By using these signal transform techniques, a vibration signal was calculated for different fuel blends, at one combustion cycle, under idle- and full-load conditions. The spectrogram (energy density spectrum) of the STFT is calculated from

$$STFT(\omega, \tau) \equiv |F(\omega, \tau)|^2 = \left| \frac{1}{\sqrt{2\pi}} \int_{-\infty}^{\infty} x(t)h(t-\tau)e^{-j\omega t}dt \right|^2. \tag{16}$$

The MSC (square of wavelet transform) is calculated from the Morlet wavelet:

$$\psi(t) = \pi^{-\frac{1}{4}} \left( e^{-i\omega_0 t} - e^{-\omega_0^2/2} \right) e^{-t^2/2}, \tag{17}$$

where $\omega_0$ ($\geq 5$) is modulation (central) frequency [68–70]. The Morlet wavelet is a complex function which can display both the amplitude and phase of the vibration signals. The windowing used in the Morlet wavelet is different from that in the STFT. The wavelet function for an input signal $x(t)$ is defined as

$$W_x(a, b; \psi) = \int x(t)\psi_{a,b}^*(t)dt, \tag{18}$$

where $a$ is a scale factor, $b$ is the wavelet windowing center or displacement along the time axis, and $\psi^*(t)$ is the complex conjugate of $\psi(t)$. Family of wavelet $\psi_{a,b}(t)$ includes a series of child wavelets, and is produced by delaying and transferring the mother wavelet $\psi(t)$:

$$\psi_{a,b}(t) = \frac{1}{\sqrt{a}} \psi \left( \frac{t-b}{a} \right). \tag{19}$$

Wavelets are performed based on a narrow window at high frequencies and a wide window at low frequencies. The MSC is then calculated as

$$SC_x(a, b; \psi) = |W_x(a, b; \psi)|^2, \tag{20}$$

where $SC_x(a, b; \psi)$ is the scalogram, and $W_x(a, b; \psi)$ is wavelet transform of $x(t)$.

In general, the STFT is more efficient for quick real-time knocks/faults detection, while complex calculations of the MSC make it more suitable for offline applications where high accuracy is required. Time-frequency representation (TFR) analysis is another powerful tool for detecting injector faults, valve defects, and pressure changes during the combustion [65].

As shown in Figure 19, the STFT and TFR analyses under the full-load condition indicated that the vibration shock was maximum when the engine was fueled with B40 and B20 blends. On the other hand, D100 and B80 fuels successfully suppressed the uncontrolled vibration during knocking. The vibration signals were represented as low- (left) and high-frequency (right) regions. As can be seen, B40 induced large engine knocking and high-frequency vibrations due to unfavorable ignition inside the cylinder and, thus, lost a large amount of the released combustion energy. Moreover, the

B20 fuel produced high amplitude vibration in the low-frequency range, indicating that most of the fuel energy was consumed for power generation at the engine. These results suggested that the type of the fuel blend has a significant influence on the efficiency of the injection system and fuel spraying.

Uludamar et al. [71] analyzed the vibration and noise (i.e., sound pressure level) performance of a direct injection diesel engine (four-stroke, four-cylinder), fueled with different volume concentrations of low sulfur diesel (D), sunflower (SB), canola (CaB), and corn (CoB) oil biodiesel fuel blends. Quality measurements of the fuels were conducted according to TS EN 14214 and EN 590 standards. All the experiments were carried out in the 1200 to 2400 rpm speed range, in a sound-insulated room and under no load condition to minimize the noise and vibration. Time- and frequency-domain analyses were performed in spatial directions. By considering the time history of the vibration wave, the RMS method was employed to find the amplitude value of the weighted and total vibration accelerations ($a_{total}$) of the engine. To provide a more accurate estimation model of the vibration and sound pressure level of the engine, other combustion properties of the test fuels (i.e., density, cetane number (CN), kinematic viscosity, and lower heating value (LHV)) were also incorporated in the calculations, using linear and non-linear regression analyses:

$$Y = \beta_0 + \beta_1 X_1 + \beta_2 X_2 + \cdots + \beta_n X_n \text{ , (linear regression)} \tag{21}$$

$$Y = a_0(X_1^{a_1})(X_2^{a_2})\cdots(X_n^{a_n}) \text{ , (non-linear regression)} \tag{22}$$

where $Y$ is a dependent variable (i.e., vibration and sound level pressure of the engine block), $\beta_i$ and $a_i$ are equation parameters, and $X_i$ are independent variables (i.e., density, CN, viscosity, and LHV).

Results revealed that with the increment of inertia forces induced by crank rotation speed, both total engine vibrations ($a_{total}$) and sound pressure level increased, on average, for all biofuels when compared to low sulfur diesel (D100). Also, the average of $a_{total}$ values of all the biodiesel blends decreased between 0.5% (CoB20) to 5.76% (CaB80), primarily due to the inherent properties of the test fuels (see Figure 20). This could be explained by the existence of higher oxygen content in the biodiesel fuels, which improves combustion quality and vibration damping of the engine block.

In a similar study, Uludamar and Yildizhan [72] also investigated the effect of aforementioned biodiesel fuel blends on vibration, noise, and exhaust emission of the same diesel engine, using the identical experimental test conditions and methodologies. However, each biofuel blend was prepared by volume ratios of only 20% and 40% (i.e., SB20, SB40, CaB20, CaB40, CoB20, and CoB40). The novelty of this work was injecting hydrogen gas ($H_2$) into intake air manifold of the engine, at flow rates of 3 and 6 min$^{-1}$, namely, H3 and H6. The energy substitution of $H_2$ is calculated from [73,74]

$$H_2 \text{ energy substitution ratio} = \frac{\dot{m}_{H_2} \times LHV_{H_2}}{\dot{m}_{fuel} \times LHV_{fuel} + \dot{m}_{H_2} \times LHV_{H_2}}, \tag{23}$$

where

$\dot{m}_{H_2} =$ the mass flow rate of $H_2$;

$\dot{m}_{fuel} =$ the mass flow rate of fuel blends;

$LHV =$ lower heating value of fuel blends.

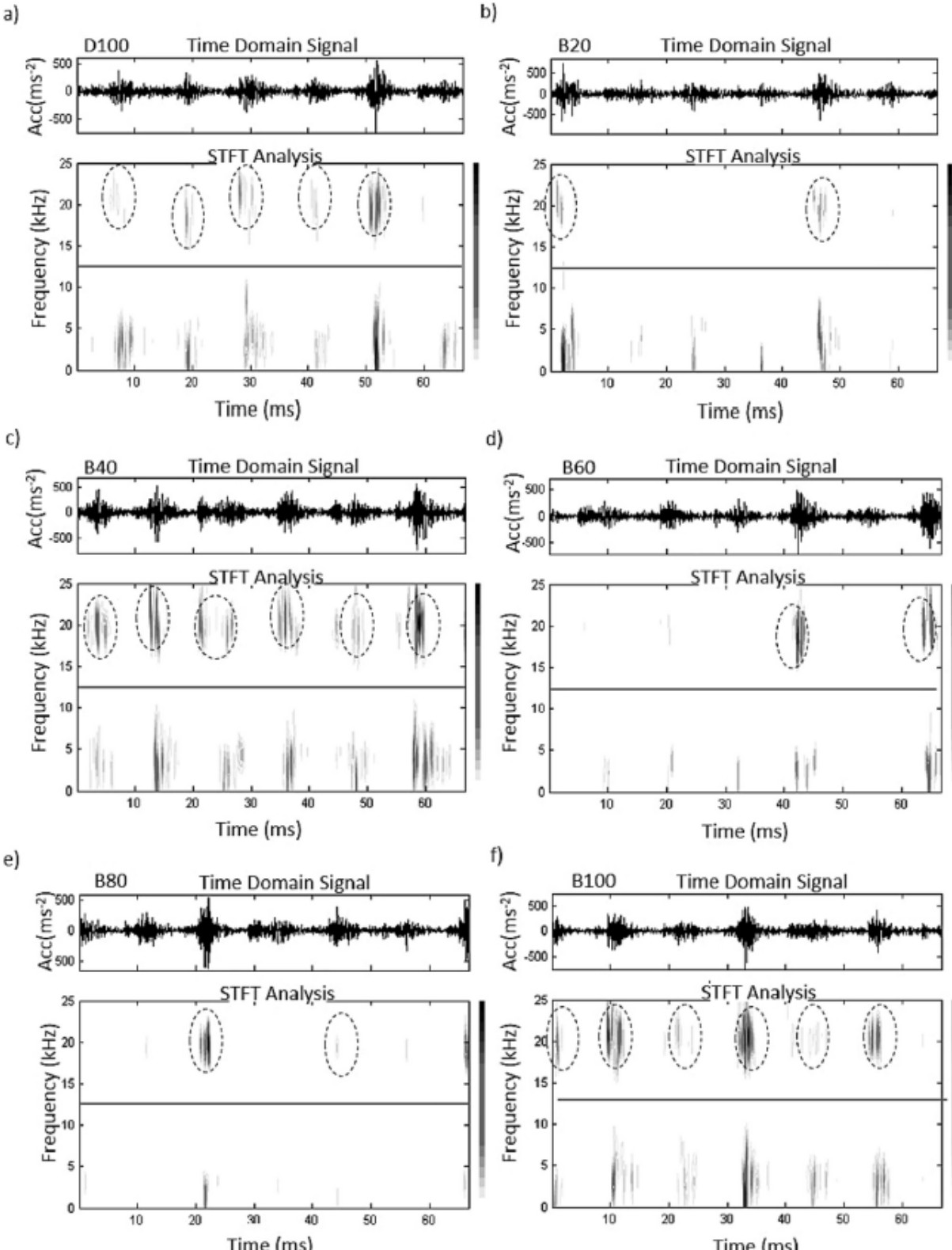

**Figure 19.** The short-time Fourier transform (STFT) analysis of vibration signals of the six-cylinder engine, running under the full-load condition at 1800 rpm, for different biodiesel–diesel fuel blends [60]. Figures are reprinted with permission from publisher.

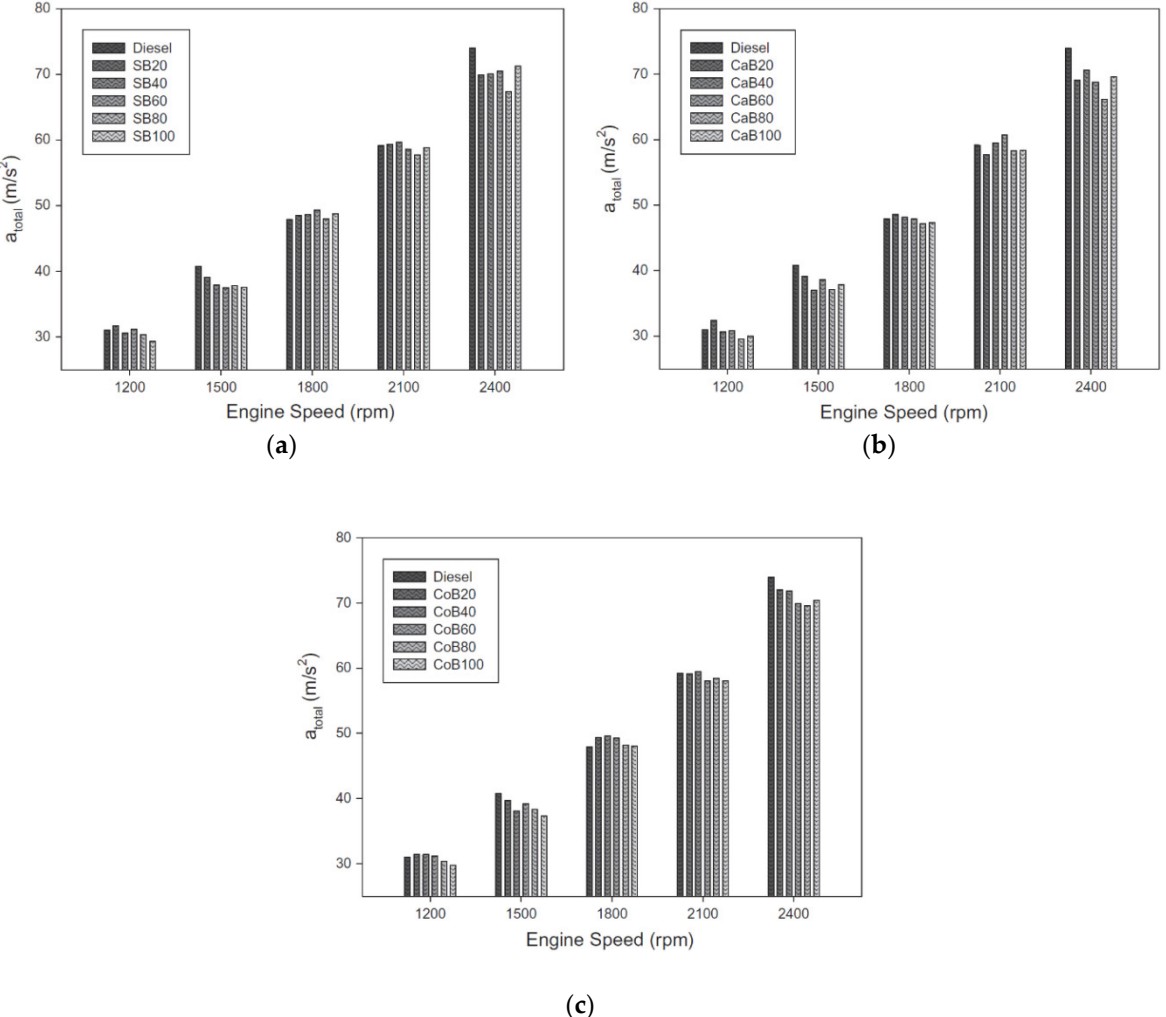

**Figure 20.** Total engine vibration acceleration ($a_{total}$) vs. engine speed when the engine was fueled with low sulfur diesel and (**a**) sunflower oil; (**b**) canola oil; (**c**) corn oil biodiesel fuel blends [71]. Figures are reprinted with permission from publisher.

The total vibration acceleration ($a_{total}$), sound pressure level, and emission values (i.e., CO, $NO_x$, and $CO_2$) of the test fuels were measured at various engine speeds (1200–2400 rpm), with and without addition of H3 and H6 to the biofuels. While the vibration acceleration decreased up to 2.79% (CaB40) with increasing the volume ratio of the biodiesel fuel blends, adding the hydrogen gases resulted in a further reduction in the vibration acceleration, up to 3.7% (SB20 + H6). These reductions were achieved mainly due to the combustion quality enhancement, ignition delay, and variation in peak pressure rise rate, after adding the hydrogen gases to the test fuels [75]. They also reported that although a higher engine speed generated more noise, substituting the conventional diesel fuel (D100) with the biodiesel fuels lowered the sound pressure level of the engine, up to 0.7 dB(A), when the hydrogen gas was mixed with the intake air. The reduction amount varied, as it is strongly dependent on combustion characteristics of each blend, such as ignition delay, peak pressure level, and pressure gradient inside the cylinders [76–78].

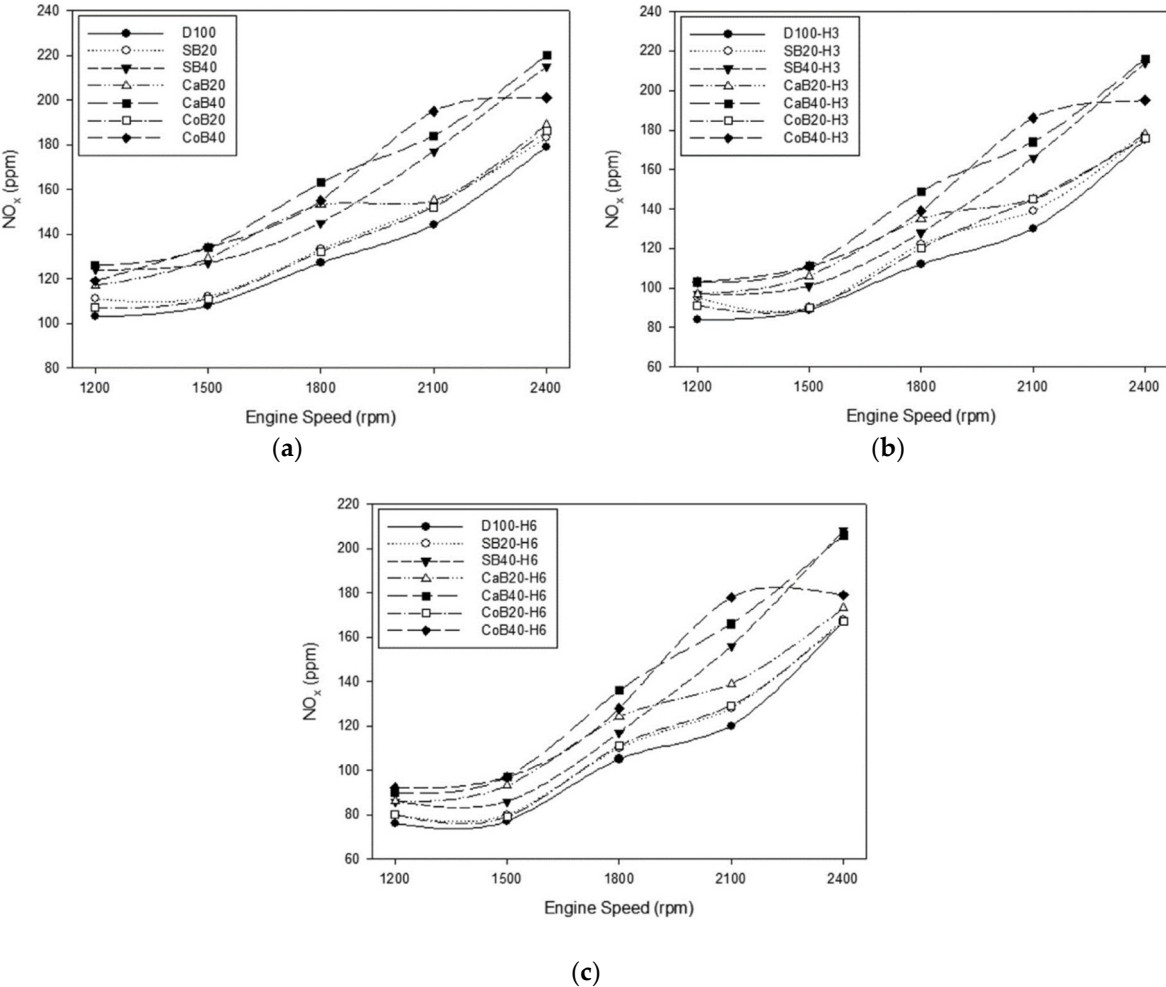

**Figure 21.** The emission values of $NO_x$: (**a**) without hydrogen; (**b**) with H3; and (**c**) with H6 addition [72]. The emission values increased at higher speeds in all test conditions. Figures are reprinted with permission from publisher.

Compared to the diesel fuel (D100), the extra amount of oxygen in the biofuels enables a complete combustion, leading to a reduction in the CO emissions [75]. However, the improvement in the combustion process provides more available carbon atoms for formation of the carbon dioxide ($CO_2$), causing an excessive increment in its emission value. This unfavorable product was fairly controlled by adding both H3 and H6 gases to the test fuels [79–81]. The other downsides with the extra oxygen content in the biodiesels in the presence of the hydrogen gas are the temperature rise during the combustion process and increase in the flame speed propagation [82]. These two factors accelerate the formation of $NO_x$, the most detrimental pollutant at the combustion stage [83]. However, those researchers observed a reduction in the $NO_x$ emissions upon addition of H3 and H6, since the atmospheric air was replaced with hydrogen gases, therefore, less nitrogen molecules flowed through the intake port (Figure 21).

Rao et al. [84] conducted a comparative study on an indirect diesel injection (IDI) engine fueled with the conventional pure diesel and mahua methyl ester (MME) oil biodiesel, blended with different alcohol-based additives. The engine contained dual combustion chambers, encouraging a lower exhaust emission and a clean combustion process. The objective of this study was to realize the efficacy of the biodiesel blends as alternatives to the conventional diesel fuel, and also investigate their impacts on the combustion pressure, heat release rate (HRR), and cylinder vibration of the four-stroke single-cylinder diesel engine. Five additives of methanol in MME were prepared with blending ratios of 1/99%, 2/98%, 3/97%, 4/96%, and 5/95%. The fuel blends were tested under five loading conditions: no load, 0.77, 1.54, 2.31, and 2.70 kW. To analyze the vibration of the cylinder

head, FFT and time waveforms were measured and recorded at each load. It was shown that proper proportions of methanol additives could reduce the heterogeneity and rapidity of the combustion process, shifting a high frequency vibration to lower frequencies with higher amplitudes, which can be successfully damped before transmitting to the base.

Taghizadeh-Alisaraei and Rezaei-Asl [85] performed a time-frequency analysis on a 6-cylinder diesel engine fueled with different bioethanol-diesel blends (by volume), to realize the effect of the ethanol concentration on combustion, knocking, and vibration of the engine. The engine was operating at 1600–2000 rpm and constant injection pressure of 35 MPa, under a full-load condition. Seven fuel blends with varying ethanol concentrations were prepared, in accordance with ASTM D5501 standard: D100 (pure diesel), D98E2 (98% diesel + 2% ethanol), D96E4, D94E6, D92E8, D90E10, and D88E12.

The RMS and spectral kurtosis (SK) are two well-adopted statistical tools in fault diagnosis of rotating machines [63,86–91], to assess energy, peakedness (sharpness), and impulsiveness of the vibration signals. While the RMS strongly depends on the acceleration amplitudes and engine speed of a system, the kurtosis is tremendously noise-sensitive and can be markedly affected by the fuel blend type and signal processing techniques (e.g., band pass filter, envelope detection). In their work, the RMS and kurtosis of the vibration (acceleration) signals were calculated to evaluate the efficacy of the engine:

$$X_{RMS} = \left[ \frac{1}{N} \sum_{k=1}^{N} x^2(t_k) \right]^{1/2}, \tag{24}$$

$$Kurt = \frac{\left[ \frac{1}{N} \sum_{k=1}^{N} x^4(t_k) \right]}{[X_{RMS}]^4}, \tag{25}$$

where $N$ is the number of points within a time period, and $x(t_k)$ is the acceleration at time of $t_k$.

Furthermore, the vibration analysis was conducted by using fast Fourier transform (FFT) and discrete STFT algorithms:

$$X(k) = \sum_{j=1}^{n} x(j) \omega_n^{(j-1)(k-1)}, \quad \omega_N = e^{(-2\pi i)/N}, \tag{26}$$

where $x(j)$ is a digital dataset with $N$ outputs $X(k)$; and

$$STFT\{x[n]\}(m, \omega) \equiv X(m, \omega) = \sum_{n=-\infty}^{+\infty} x[n] h[n-m] e^{-j\omega n}. \tag{27}$$

Herein, $x[n]$ is the discrete-time signal to be transformed, $x[n]h[n-m]$ is a short-time section of the vibration signal at time $n$ with the length of $m$ and window function of $h[n]$; and $X(m, \omega)$ is the FFT of each section. Results showed that several factors contribute to the engine vibration, including the fuel uniformity, oxygen solubility, combustion process, fuel injection, fuel powdering, fuel vaporization, and ignition delay. The last four are significantly influenced by the ethanol content in the fuel and can cause engine knocking at high concentrations (e.g., D90E10 and D88E12). Among the studied fuels, D94E6 blend generated more power and torque, while D92E8 enhanced pressure changes through the knocking phenomenon (generating shocks) within the cylinders (see Figure 22). The engine power (*P*) was calculated from the available torque (*T*) at the engine flywheel and engine speed (*n*):

$$P = \frac{2\pi T n}{60,000} = \frac{T n}{9554}. \tag{28}$$

As illustrated in Figure 22, a further increase in the engine speed after 1900 rpm (maximum power) resulted in a considerable reduction in the generated torque and power of all fuel blends.

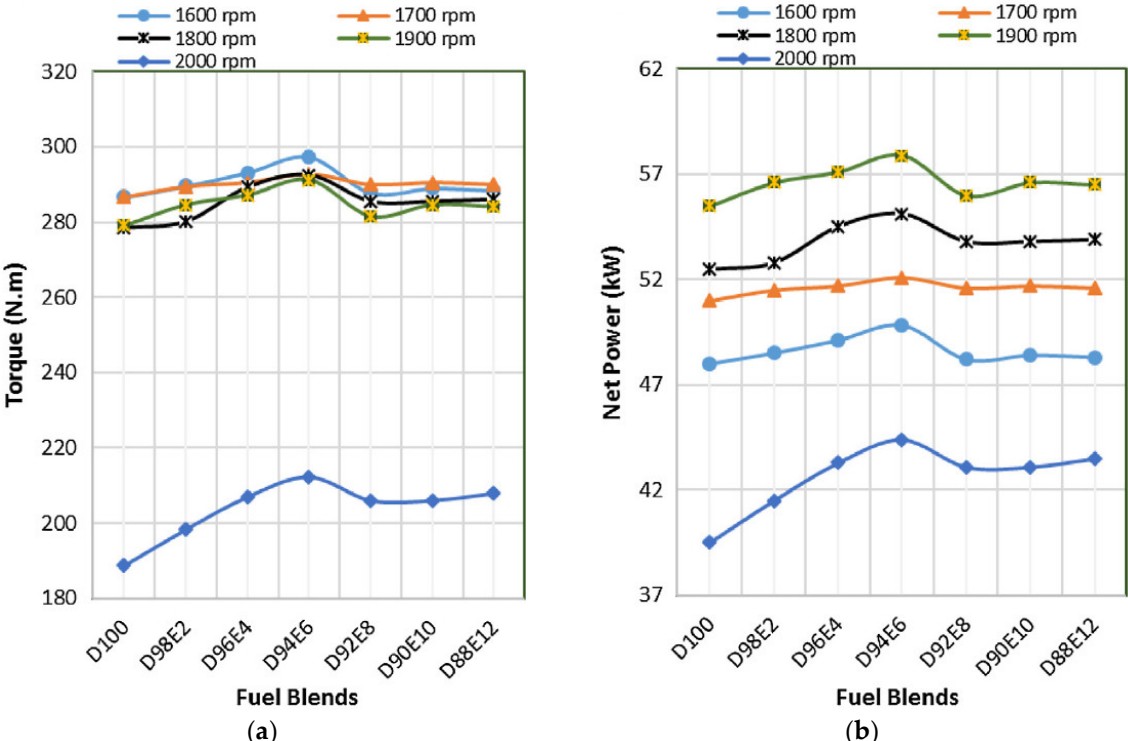

**Figure 22.** The generated (**a**) torque and (**b**) power of the ethanol/diesel fuel blends, at the speeds of 1600−2000 rpm. [85]. Figures are reprinted with permission from publisher.

Despite their promises, the engine performance was very erratic due to the increase in the unwanted acceleration of the body, and an ignition delay of D94E6, D92E8 fuel blends, respectively (Figure 23).

Ravi et al. [92] conducted a comparative study on the engine performance and vibration of different fuel blends. Their results demonstrated that, in the diesel engine, vibrations started to rise, whereas in biodiesel and diesel blends (B10 to B30) it declined. This may be due to the reduction in the knocking effect in biodiesel, as well as diesel blends' combustion. As a result of good combustion, the vibration decreases at higher blends of biodiesels.

In a recent study, Javed et al. [93] investigated the effect of adding zinc oxide (ZnO) nanoparticles with hydrogen in dual fuel mode to jatropha methyl ester (JME) biodiesel on the engine vibration. The main results of this study are listed as follows: Firstly, increasing nanoparticle size improved the combustion characteristics, which therefore results in significant improvement in vibration behavior. Secondly, among the fuel blends, B30JME40 (30% JME, 70% diesel, 40 nm size nanoparticle) and B20JME40 (20% JME, 80% diesel, 40 nm size nanoparticle) displayed a better performance due to having high heat release rate (HRR), metal vaporization and oxidation, catalytic behavior, and combustion rate, which generated lower vibrations (under all loads, when $H_2$ gas was added at 0.5 L min$^{-1}$ to the blends.

In an experimental study carried out by Patel et al. [94], the vibration and performance of a single cylinder IC engine fueled with mineral diesel, jatropha biodiesel (JB100), and jatropha biodiesel–diesel blend (JB20), were thoroughly compared. The results showed that the engine fueled with JB100 generated the least vibrations. On the other hand, the JB20 fuel displayed 7–25% more vibration than the mineral diesel fuel in the vertical direction, while the vibration for the JB100 fuel was 1–25% lower than the mineral diesel.

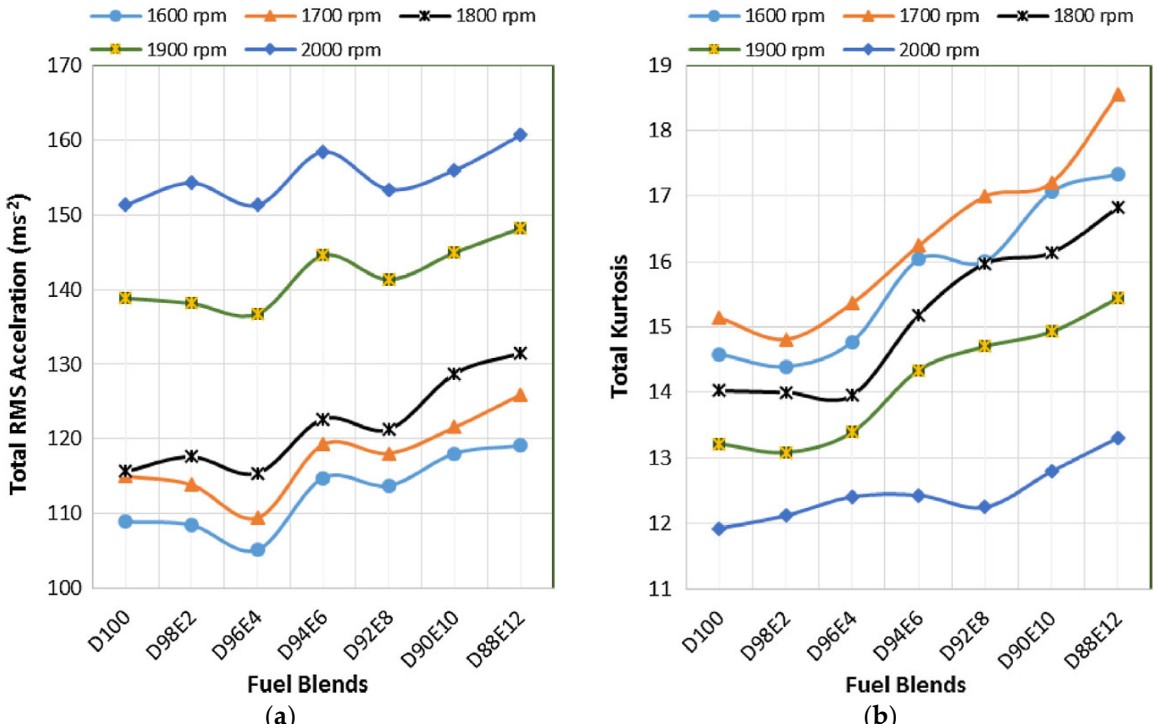

**Figure 23.** (**a**) Resultant acceleration diagram for different fuel blends; (**b**) resultant kurtosis chart for different fuel blends [85]. Figures are reprinted with permission from publisher.

In another experimental study carried out by Patel et al. [95], the same investigation was done with a mineral diesel, Karanja biodiesel (KB100), and Karanja biodiesel–diesel blend (KB20). The results showed that the diesel combustion exhibited the best vibration performance in the lateral direction, whereas KB100 generated the highest levels of vibrations in this direction. However, in the piston direction, KB20 generated the maximum level of vibrations, while KB100 generated the least vibrations. Moreover, since the amplitude of the vibration in the piston direction was 2.5 to 3 times more than that of other directions, it was concluded that KB100 had the best overall behavior in the vibratory motion.

Lately, Liu et al. [96] comprehensively reviewed the most recent biodiesel fuels, developed by using ethylene glycol monomethyl ether, and reported their effects on engine combustion and emissions. They compared various ether group alcohols to fatty acid methyl ether (FAME,) including ethylene glycol monomethyl ether, ethylene glycol monoethyl ether, ethylene glycol monopropyl ether, propylene glycol monomethyl ether, diethylene glycol monomethyl ether, and triethylene glycol monomethyl ether. The major advantage of using the new biofuels is proliferation of the oxygen content in their ether groups, which increases their CN (over 70) and reduces particular matter (PM) formation and emissions. This elevated CN shortens the ignition delay time, resulting in lower $NO_x$ formation and emissions, and enhanced the biodiesel fuel combustion.

## 5. Conclusion

This review paper aimed to provide a comprehensive overview and sound summary of all the work undertaken recently, addressing the variety of methods and techniques developed for mitigating or reducing vibration, noise, and emissions (e.g., CO, $CO_2$, $NO_x$), mainly in diesel and direct injection engines. The conventional statistical tools and mathematical approaches that are commonly used in vibration diagnosis of the ICEs, such as root mean square (RMS), fast Fourier transform (FFT), short-time Fourier transform (STFT), spectral kurtosis (SK), are reviewed thoroughly. In particular, the paper elaborated on the efficient modifications that can be applied to the engine or potential use of innovative biofuels that can be used to substitute diesel in ICEs. The engine modifications were applied through synthesizing novel mechanisms and changing the

design or geometry of crucial engine components. On the other hand, adding a wide range of additives to the conventional diesel fuel, or altering the properties of current biodiesel–diesel fuel blends, such as cetane number (CN), lower heating value (LHV), density, viscosity, etc., enables commercializing more reliable and environmentally friendly fuels with lower noise and chemical pollutions. The fuel additives discussed in this article were vegetable oils (sunflower, canola, corn), alcohols (methanol, ethanol), hydrogen gas, zinc oxide (ZnO) nanoparticles, and ethylene glycol monomethyl ether.

The most important conclusions reached from the literature review are:

- Introducing new balancing algorithms accompanied by time- and frequency-domain analyses enables reduction of the torsional vibration in rotary components of ICEs operating under high dynamic loading conditions.
- By proper use of engine mounting system (EMS), when a high transmission ratio is selected, the vibration and noise can be remarkably isolated before they are transferred to chassis and body of a vehicle.
- Accurate prediction of time and frequencies of the piston slap events during the engine cycle can be achieved by taking into account the impact of elastohydrodynamic lubrication (EHL) and piston transient dynamics.
- The torsional vibration of ICEs can be significantly mitigated by optimizing the geometrical parameters of the engine's crank-slider mechanism and designing a novel kinematically driven flywheel (KDF).
- To minimize the transmitting vibrations, the stiffness of the engine block can be enhanced by applying structural modifications such as thickening the crankcase walls, front gear cover, back flywheel cover, as well as adding ribs to its driveshaft.
- The resistive torque fluctuations of an engine camshaft can be passively compensated by carefully designing a balancing cam mechanism.
- RMS, kurtosis, STFT, FFT, and Morlet wavelet are among the most common statistical tools that are used for vibration and noise analyses in ICEs.
- A biodiesel fuel (or biofuel) offers better physicochemical properties (i.e., density, viscosity, distillation, bulk modulus, surface tension, etc.), to control the injection process.
- Canola and corn vegetable oil fuel additives can avoid an abrupt combustion process due to increasing the oxygen content in the biodiesel fuels.
- When hydrogen gas is added to the fuel blend, it improves the combustion quality, ignition delay, and variation in peak pressure rise rate, resulting in a reduction of the noise and total vibration of the engine.
- The vibration performance of ICEs can be significantly enhanced by adding ZnO nanoparticles to biofuels, as they have a higher heat release rate (HRR), metal vaporization and oxidation, catalytic behavior, and combustion rate.
- Finally, fatty acid methyl ether (FAME) biofuels containing ether group alcohols can reduce the ignition delay time and $NO_x$ pollutants, leading to better biodiesel fuel combustion.

**Author Contributions:** All authors contributed equally.

**Funding:** This research received no external funding.

**Conflicts of Interest:** The authors declare no conflict of interest.

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
