# Peer review of "Performance Enhancement of Internal Combustion Engines through Vibration Control: State of the Art and Challenges"

_applsci, doi:10.3390/app9030406_

Round 1

Reviewer 1 Report

The submitted review paper describes various methods and techniques for vibration control in internal combustion engines. The subject of the paper is quite interesting and it can be proven quite useful to scientists and engineers working on solutions for vibration control and mitigation in internal combustion engines. Authors performed a very detailed literature review and the conclusions of the paper are justified by the analysis of the paper manuscript. I have only one objection to the organization of the paper manuscript: According to my opinion, it would be more beneficial for the technical integrity and the long-term visibility of the paper if authors devoted a separate paragraph after "Introduction", which will present the basic theory and mathematical equations of the vibration generation and processing. I believe a brief description of the theory of vibrations and their processing methods in a separate paragraph apart from the examination of specific case studies will enhance the technical integrity and the long-term validity of the paper.

Author Response

We would like to sincerely thank the reviewers and editors for contributing to the improvement of our manuscript’s quality. The manuscript was reviewed for clarity, diction, and to ensure the work is easily and effectively conveyed to the reader. Also, we have tried with our best effort to address all of the comments and to answer any question raised by the reviewers. With the revisions outlined in this letter, we hope the manuscript can be considered for publication.

Reviewer 1:

1. According to my opinion, it would be more beneficial for the technical integrity and the long-term visibility of the paper if authors devoted a separate paragraph after "Introduction", which will present the basic theory and mathematical equations of the vibration generation and processing. I believe a brief description of the theory of vibrations and their processing methods in a separate paragraph apart from the examination of specific case studies will enhance the technical integrity and the long-term validity of the paper.

We would like to thank the reviewer for the positive remarks regarding our manuscript; it is greatly appreciated. We added a small section to the current revised version of the manuscript, describing the basics of vibration and signal processing. More complex concepts and techniques were discussed within the text, wherever needed. Please see section 2, page 2, of the revised manuscript.

Reviewer 2 Report

The review paper is well organized  and well written.  The authors present both experimental as well as numerical simulation results. In general, the deterioration of the engine performance during running conditions is a crucial problem, especially in the sector of transportation vehicles and the authors made a significant effort to come up with a review report. The manuscript comprises an amount of interesting findinds; however, they are presented in a way that does not emphasize their importance. Thus, the manuscript is in principle suitable for publication in Applied Sciences, but only after minor revisions implemented by the authors in the following points.

In section 2 the vibration signals should be described clearly. It is important to note that there are a host of parametrs such as combustion, piston sLap, pisto ring-,liner contact, valve events etc which can affect vibration and noise taken on engine block. Deeper discusion is necessary in this review work. The following original articles can help your information:

Zavos A and Nikolakopoulos PG. Measurement of frictio n and noise from piston assembly of a single-cylinder motorbike engine at realistic speeds, Part D; Journal of Automobile Engineering , 2018.

Dolatabadi N et al. A transient tribodynamic approach for rhe calculation of internal combustion engine piston slap noise, Journal of Sound and Vibration, 2015.

In section 3.3, the behavior of the engine oil is an important matter for anyone dealing with vibration and noise;however, the authors  do not make any comments about viscosity  dependence on the temperature and pressure through this manuscript. 

page 19, lines 363-365: the authors call the STFT about knots/faults detection. Where does this come from. Add more references.

conclusions: A better re-arrangement of the information would reveal the significance of this review work. The impact and the next plan of this topic should be given.

Author Response

Reviewer 2:

1. The review paper is well organized and written.  The authors present both experimental as well as numerical simulation results. In general, the deterioration of the engine performance during running conditions is a crucial problem, especially in the sector of transportation vehicles and the authors made a significant effort to come up with a review report. The manuscript comprises an amount of interesting findings; however, they are presented in a way that does not emphasize their importance. Thus, the manuscript is in principle suitable for publication in Applied Sciences, but only after minor revisions implemented by the authors in the following points.

We would like to thank the reviewer for the positive remarks regarding our manuscript.

2. In section 2 the vibration signals should be described clearly. It is important to note that there are a host of parameters such as combustion, piston slap, piston ring-liner contact, valve events etc. which can affect vibration and noise taken on engine block. Deeper discussion is necessary in this review work.

Thank you for your comments. We agree with the review on this. A deeper discussion and proper citations were added to the current version of the manuscript. Please see page 8, lines 194-210, in the current revised manuscript.

3. In section 3.3, the behavior of the engine oil is an important matter for anyone dealing with vibration and noise; however, the authors do not make any comments about viscosity dependence on the temperature and pressure through this manuscript.

We thank the reviewer for bringing this to our attention. We agree with the review on this. A deeper discussion and proper citations were added to the current version of the manuscript. Please see page 14, lines 321-325, in the current revised manuscript.

4. Page 19, lines 363-365: the authors call the STFT about knots/faults detection. Where does this come from. Add more references.

We thank the reviewer for bringing this to our attention. Proper citations were added to the current version of manuscript. Please see refs. 60-66, page 18, line 432.

5. Conclusions: A better re-arrangement of the information would reveal the significance of this review work. The impact and the next plan of this topic should be given.

We agree that a better rearrangement will enhance the visibility of the work. Therefore, a separate section was devoted to the mathematical modeling of torsional vibration in ICEs, and proper citations were added. The contribution of this paper as well as the future plan were provided in the conclusion section. Please see section 2, page 2, and the conclusion.

Reviewer 3 Report

The paper is a comprehensive review on methods for vibration control in internal combustion engines.

The work is well organized and written. The state of the art is quite thorough, though it can be furtherly improved (see for instance "Guo et al., Diesel engine torsional vibration control coupling with speed control system, Mechanical Systems and Signal processing 94 (2017), 1-13).

Minor remarks: please improve the overall quality of the figures (and, when possible, enhance the font size of the legends)

Author Response

Reviewer 3:

1. The paper is a comprehensive review on methods for vibration control in internal combustion engines. The work is well organized and written. The state of the art is quite thorough, though it can be furtherly improved.

We would like to thank the reviewer for the positive remarks regarding our manuscript. A deeper discussion and proper citations were added to the current version of the manuscript. Please see section 2, page 2, in the revised manuscript.

2. Please improve the overall quality of the figures (and, when possible, enhance the font size of the legends).

We thank the reviewer for bringing this to our attention. The quality of the pictures was enhanced and the legend font size also improved.

Round 2

Reviewer 1 Report

Authors performed all required revisions, improve substantially the technical and the editorial quality of their paper manuscript and addressed all issues pointed out by the reviewers. Hence, their revised paper manuscript can be published without further modifications.

Reviewer 2 Report

The revised paper is acceptable.